# Protrudin functions from the endoplasmic reticulum to support axon regeneration in the adult CNS

Veselina Petrova [1✉], Craig S. Pearson[2], Jared Ching[1,3,4], James R. Tribble[5], Andrea G. Solano[2], Yunfei Yang [4], Fiona M. Love [1], Robert J. Watt[1], Andrew Osborne[1], Evan Reid [6], Pete A. Williams [5], Keith R. Martin [1,7,8], Herbert M. Geller [2], Richard Eva [1,10✉] & James W. Fawcett [1,9,10✉]

Adult mammalian central nervous system axons have intrinsically poor regenerative capacity, so axonal injury has permanent consequences. One approach to enhancing regeneration is to increase the axonal supply of growth molecules and organelles. We achieved this by expressing the adaptor molecule Protrudin which is normally found at low levels in non-regenerative neurons. Elevated Protrudin expression enabled robust central nervous system regeneration both in vitro in primary cortical neurons and in vivo in the injured adult optic nerve. Protrudin overexpression facilitated the accumulation of endoplasmic reticulum, integrins and Rab11 endosomes in the distal axon, whilst removing Protrudin's endoplasmic reticulum localization, kinesin-binding or phosphoinositide-binding properties abrogated the regenerative effects. These results demonstrate that Protrudin promotes regeneration by functioning as a scaffold to link axonal organelles, motors and membranes, establishing important roles for these cellular components in mediating regeneration in the adult central nervous system.

[1] John van Geest Centre for Brain Repair, Department of Clinical Neurosciences, University of Cambridge, Cambridge, UK. [2] Laboratory of Developmental Neurobiology, Division of Intramural Research, National Heart, Lung and Blood Institute, NIH, Bethesda, USA. [3] MRC Mitochondrial Biology Unit, University of Cambridge, Cambridge, UK. [4] Department of Ophthalmology, Addenbrooke's Hospital, Cambridge, UK. [5] Department of Clinical Neuroscience, Division of Eye and Vision, St. Erik Eye Hospital, Karolinska Institutet, Stockholm, Sweden. [6] Cambridge Institute for Medical Research and Department of Medical Genetics, University of Cambridge, Cambridge, UK. [7] Ophthalmology, Department of Surgery, University of Melbourne, Melbourne, Australia. [8] Centre for Eye Research Australia, Royal Victorian Eye and Ear Hospital, Melbourne, Australia. [9] Centre for Reconstructive Neuroscience, Institute of Experimental Medicine CAS, Prague, Czech Republic. [10] These authors contributed equally: Richard Eva, James W. Fawcett. ✉email: vp351@cam.ac.uk; re263@cam.ac.uk; jf108@cam.ac.uk

Axons of immature central nervous system (CNS) and adult peripheral nervous system (PNS) neurons readily regenerate after injury[1,2]. In contrast, adult CNS neurons lose their regenerative ability with maturation[3], meaning that axonal injury or disease has life-altering consequences and that there is little chance of recovery. In addition to the non-permissive extracellular environment after injury, intrinsic neuronal factors also play an important role in the regenerative failure observed in mature CNS neurons[4,5]. Studies aimed at enhancing CNS regeneration have identified transcriptional and epigenetic programs[6,7], signalling pathways[8–10], the cytoskeleton[11–14] and axon transport[15–19] as important factors governing regenerative ability. However, the precise machinery required to reconstitute and extend an injured axon is not completely understood, and repairing the injured CNS remains a challenging objective[20].

This study focuses on the adaptor molecule Protrudin as a tool for investigating and enhancing axon growth and regeneration in the adult CNS. Protrudin is an integral endoplasmic reticulum (ER) membrane protein that has two properties which make it a candidate for enabling axon regeneration. First, overexpression of Protrudin causes protrusion formation in non-neuronal cell lines, and promotes neurite outgrowth in neuronal cells[21]; second, Protrudin is a scaffolding molecule which possesses interaction sites for key axon growth-related molecules and structures[21]. Through its Rab11 and kinesin-1 (KIF5) binding sites, Protrudin can enable the anterograde transport of Rab11-positive recycling endosomes, leading to their and their cargo's accumulation at protrusion tips[21,22]. This is relevant to CNS axon repair because increased Rab11 transport into CNS axons in vitro increases their regenerative ability[19].

Protrudin localizes to the ER through two transmembrane domains and a hairpin loop and interacts with VAP proteins at ER-membrane contact sites through an FFAT domain. This interaction is involved in its effects on protrusion outgrowth[23–25]. In addition to its localization at ER tubules, Protrudin also regulates ER distribution and network formation[23]. Protrudin also has a FYVE domain that binds to phosphoinositides enabling interaction with endosomes and the surface membrane[25]. Specific phosphoinositides are required at the growth cone during rapid axon growth and for axon regeneration[26]. Protrudin therefore links a number of cellular components associated with axonal growth. Our hypothesis was that the expression of an active form of Protrudin would enable regeneration of CNS axons via a combination of these interactions and would be a powerful tool for understanding the mechanisms of axon regeneration.

Our initial studies found that Protrudin mRNA is expressed at low levels in CNS neurons, but at higher levels in regenerating PNS neurons, and the protein is present in immature regenerative CNS axons but is depleted from axons with maturity as regeneration is lost. We reasoned that overexpression might allow for increased availability of regenerative machinery within the axon, leading to better regeneration after injury. Because stimulation of protrusions through the interaction of Protrudin and Rab11 is increased by growth factor receptor phosphorylation, and its protrusive effects are prevented by dominant-negative mutations of the ERK phosphorylation sites, we created a phosphomimetic, active form of Protrudin, mutating these previously identified phosphorylation sites[21].

Here, we report that Protrudin promotes axon regeneration through the mobilization of endosomes and ER into the distal part of injured axons, whilst having striking effects on neuronal survival after axotomy. Importantly, Protrudin expression only has moderate effects on developmental axon growth but has strong effects on regeneration. Overexpression of Protrudin, particularly in its phosphomimetic form, led to robust regeneration and neuroprotection of mature cortical axons in vitro and

of retinal ganglion cell (RGC) axons in the injured optic nerve. Protrudin expression caused increased transport of Rab11 endosomes, integrins, and an accumulation of ER in the axon tip, with phosphomimetic Protrudin increasing this effect. Deleting either the ER transmembrane domains or VAP-binding FFAT domain prevented the accumulation of the ER whilst also abrogating the effects on regeneration. Interfering with other key domains of Protrudin also eliminated the regenerative effects, indicating that Protrudin promotes regeneration by acting as a scaffold in the axonal ER, bringing together growth components, organelles and membranes to enable CNS axon regeneration.

## Results
Protrudin has several key domains, including a Rab11-binding domain (RBD), three hydrophobic membrane-association domains (TM 1–3), an FFAT motif for binding to VAP proteins at ER contact sites, a coiled-coiled (CC) domain (which interacts with kinesin 1) and a phosphoinositide-binding FYVE domain which enables interaction with endosomes and the plasma membrane[21,23–25,27]. The map of these domains is shown in Fig. 1a. Figure 1b shows the phosphorylation sites that were mutated to produce phosphomimetic Protrudin in accordance with the previous literature[21]. We tested the ability of wild-type and phosphomimetic Protrudin to induce protrusion formation in HeLa cells as previously described as a distinctive feature of Protrudin[21]. We found that both wild-type and phosphomimetic Protrudin stimulate protrusion formation compared to cells expressing a control construct (Supplementary Fig. 1a, b). Cells overexpressing phosphomimetic Protrudin showed a higher percentage of protrusion formation as well as longer protrusions compared to wild-type Protrudin confirming the active phenotype of our phosphomimetic form of the protein (Supplementary Fig. 1a–c).

**Protrudin is expressed at low levels in mature, non-regenerative CNS neurons**. In order to affect growth and regeneration, we reasoned that Protrudin would need to be present in axons in a significant quantity and be optimally functional. We first examined the mRNA expression of Protrudin in developing CNS and PNS neurons, as well in PNS neurons after injury, using previously published RNA-sequencing datasets. We found that Protrudin mRNA (Zfyve27) is expressed at low levels in CNS neurons, and its expression is not developmentally regulated (Fig. 1c)[19]. In contrast, in sensory, regeneration-capable neurons, the Protrudin transcript increases with development, during axon growth in vitro, and in response to peripheral nerve injury (Fig. 1d)[28].

To assess the level and distribution of Protrudin protein in CNS axons, we examined its endogenous localization in rat primary cortical neurons by immunocytochemistry. We compared developing neurons (2–4 days after plating at E18), with mature neurons that have lost the ability to regenerate their axons (matured in vitro for 14+ days). We found that Protrudin localized to both axons and dendrites of developing cortical neurons but was enriched in dendrites and restricted from axons at later stages, coinciding with the time when cortical neurons mature and lose their regenerative ability (Fig. 1e, f)[19]. By measuring the axon and dendrite fluorescence intensity of Protrudin immunolabelling we found that younger neurons (2–4 days in vitro, DIV) had a higher axon-to-dendrite ratio (ratio = 1) compared with later stages of development (7–9 DIV, ratio = 0.47 and 14–16 DIV, ratio = 0.44) (Fig. 1g). These observations suggest that endogenous Protrudin may not be present in sufficient quantity in mature CNS axons to influence regeneration. Overexpression of either wild-type or phosphomimetic Protrudin resulted in a substantial increase in the protein level in rat primary cortical neurons (Supplementary Fig. 2). This

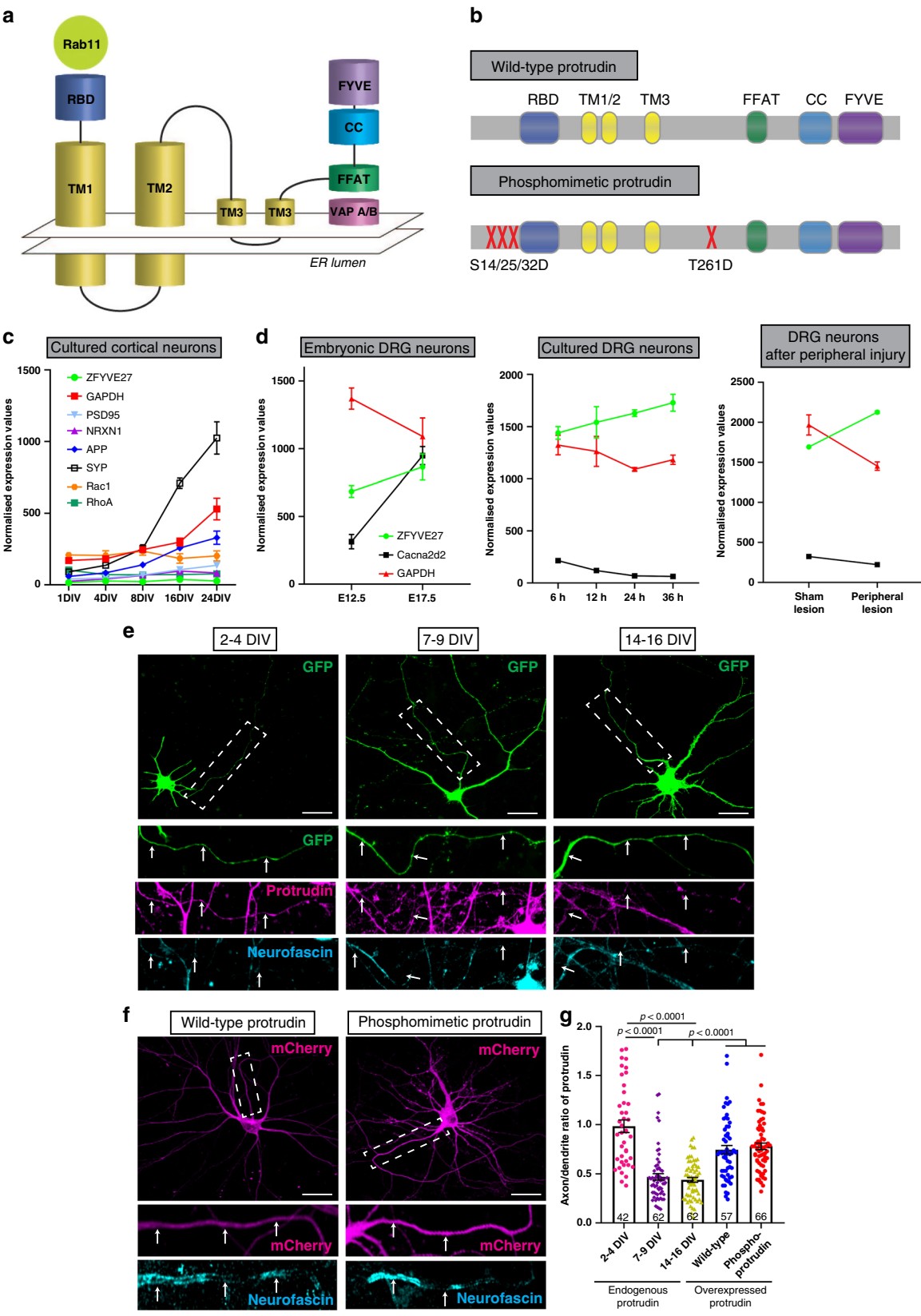

resulted in an increased axon-to dendrite ratio in neurons overexpressing wild-type (ratio = 0.77) or phosphomimetic (ratio = 0.78) Protrudin at 14–16 DIV indicating an increase in the protein's distribution to axons, with Protrudin easily detectable throughout axons (Fig. 1f, g). The exclusion of Protrudin from mature axons is therefore not absolute and can be overcome by overexpression. Overexpression of Protrudin had no effect on soma size or spine number and morphology whilst phosphomimetic Protrudin had a modest effect on increasing dendritic tree complexity (Supplementary Fig. 3).

**Fig. 1 Protrudin is expressed at low levels in mature axons and overexpression restores this deficit. a** Schematic diagram of Protrudin's domains and structure. **b** Schematic of wild-type and phosphomimetic Protrudin mutagenesis sites. **c** mRNA expression levels of six neuronal genes (including *Zfyve27*— the Protrudin gene) from different stages of development in primary rat cortical neurons in vitro (*n* = 5–6 samples). **d** Normalized expression levels of Protrudin and other related genes during embryonic development in the mouse (*n* = 3 animals for each timepoint), after plating DRG neurons in vitro (*n* = 3 animals for each timepoint) or after peripheral nerve injury in DRG cells (*n* = 3 animals for sham and injured samples). **e** Immunofluorescent images of Protrudin in the proximal axons (white dotted line box) of neurons at different stages of development in culture. Scale bars are 20 μm. The white arrows follow the course of the proximal axon. **f** Immunofluorescent images of overexpressed, mCherry-tagged wild-type or phosphomimetic Protrudin (magenta) and staining for the axon initial segment marker—neurofascin (cyan). Scale bars are 20 μm. **g** The axon-to-dendrite ratio of Protrudin at different developmental stages or after overexpression at 14–16 DIV (*n* = 4 independent experiments for each condition, *n* numbers on graph show the number of analysed cells) (Kruskal–Wallis with Dunn's multiple comparison test, *p* < 0.0001, Kruskal–Wallis statistic = 101.6). Error bars represent mean ± SEM.

**Overexpression of wild-type or phosphomimetic Protrudin enhances axon regeneration in vitro**. Protrudin overexpression has previously been associated with enhanced neurite outgrowth in HeLa and PC12 cells as well as primary hippocampal neurons at early stages of development[21]. Given that Protrudin is expressed at low levels in CNS neurons, we reasoned that its overexpression might increase axon growth. We transfected wild-type or phosphomimetic Protrudin into immature 2 DIV cortical neurons and measured the growth of axons and dendrites at 4 DIV (Fig. 2a). Overexpression of phosphomimetic Protrudin had a modest effect on this early phase of axon growth, with neurons overexpressing phosphomimetic Protrudin having increased length of the longest neurite (580 μm) compared to control-transfected neurons (470 μm) (Fig. 2b). Overexpression of wild-type Protrudin had no effect. These results show that high levels of phosphomimetic Protrudin have a small effect on the initial outgrowth phase of neurons in culture. At this time axon growth is already rapid (i.e. axons usually regenerate when cut), and the maturation-related compartmentalization of the neurons to exclude growth-related molecules from axons has not yet occurred[19].

We next examined the effect of Protrudin overexpression on axon regeneration in mature axons. We used the laser axotomy model of regenerative decline, where axons progressively lose their regenerative ability as they mature and become electrically active (Fig. 2c)[18,19]. Cortical neurons were again transfected with either control, wild-type or phosphomimetic Protrudin, this time at 10 DIV. We examined axon regeneration after laser injury at 13–17 DIV, when regenerative capacity has declined[19]. In this model, axons typically show two responses to injury, either the formation of a retraction bulb and no regeneration, or retraction and bulb formation followed by growth cone development and axon extension (Fig. 2d). Expression of either wild-type or phosphomimetic Protrudin led to a dramatic increase in the percentage of axons regenerating after laser axotomy (Fig. 2e, Supplementary Video 1) with axons regenerating longer distances (Fig. 2f) and initiating regeneration in a shorter time (Fig. 2g). The speed of axon extension after growth cone initiation did not differ between the three conditions (Fig. 2h). This indicates that Protrudin has its most pronounced effect on initial growth cone formation, rather than on the axon elongation phase of regeneration. These regenerative events were most pronounced in neurons transfected with phosphomimetic Protrudin. Importantly, overexpressed wild-type and phosphomimetic Protrudin was found to localize throughout axons, accumulating at the growth cones of uninjured axons, at regenerating growth cones, and at the retraction bulbs of non-regenerating injured axons. At the growth cone, Protrudin localized principally to the central domain (Supplementary Fig. 4d–f).

Protrudin's effect on axon regeneration was dose dependent; co-transfection with a construct encoding GFP resulted in lower Protrudin expression and a reduced effect on regeneration (Supplementary Fig. 4a–c). These results show that Protrudin,

particularly in its constitutively active phosphomimetic form, has a very strong effect on the rescue of axon regeneration in mature neurons. There is therefore a contrast between Protrudin's minor enhancement of the outgrowth of immature axons that are already growing rapidly, and the rescue of regeneration in mature neurons whose axons seldom regenerate.

**Protrudin promotes regeneration through increased endosomal transport**. Protrudin has several interaction sites that have the potential to link endosomes, ER, membrane and kinesin. Our hypothesis, based on studies of how Protrudin causes neurite growth in HeLa and PC12 cells[21] and our studies of Rab11 vesicles and their cargo in regeneration[19], was that the main effect of Protrudin would be to enable transport of Rab11 vesicles and their contents into mature axons through linkage to KIF5, so increasing regenerative capacity. In order to determine if Protrudin's regenerative effects were mediated through enhanced axonal transport, we examined the transport of Rab11 in the presence of wild-type or phosphomimetic Protrudin, and also studied the transport of a known Rab11 cargo, integrin alpha 9[29]. This integrin can mediate long-range sensory regeneration in the spinal cord[15]. In addition, we overexpressed three mutated Protrudin constructs targeting domains associated with endosomal transport (Rab-binding domain, KIF5-interaction domain, and FYVE domain), and examined axon regeneration after laser axotomy.

To determine whether Protrudin's regenerative effects were also accompanied by an increase in axonal transport, we used spinning-disc live-cell microscopy to observe the movement of Rab11-GFP or integrin α9-GFP in the distal part of mature, 13–17 DIV axons in the presence of overexpressed wild-type or phosphomimetic Protrudin (Fig. 3a). Vesicle transport was scored as anterograde, retrograde, bidirectional or static and the total number of Rab11 or integrin α9-positive endosomes per section of axon was measured. The majority of Rab11-positive vesicles trafficked bidirectionally whereas the bulk of integrin-containing endosomes moved retrogradely confirming previous studies[18,29]. Overexpression of either wild-type or phosphomimetic Protrudin resulted in increased retrograde and bidirectional transport of Rab11-GFP and enhanced anterograde and retrograde transport of integrin α9-GFP (Fig. 3b–d), leading to more total Rab11 and integrin-positive vesicles in the distal axon (Fig. 3b–d). The finding that phosphomimetic Protrudin has no additional effect on axonal transport compared with wild-type Protrudin suggests that phospho-mimetic Protrudin does not function to further stimulate Rab11 transport. Approximately 20% of α9-transporting vesicles were positive for Protrudin (Supplementary Fig. 5a, b) and 29% of Protrudin-positive endosomes (wild-type or phosphomimetic) were also Rab11-positive (Supplementary Fig. 5a, b), and kymograph analysis demonstrated dynamic co-localization of both Rab11 and α9-integrin with Protrudin.

To test the contribution of Protrudin's transport-associated domains towards its regenerative effects, we assembled a cohort of

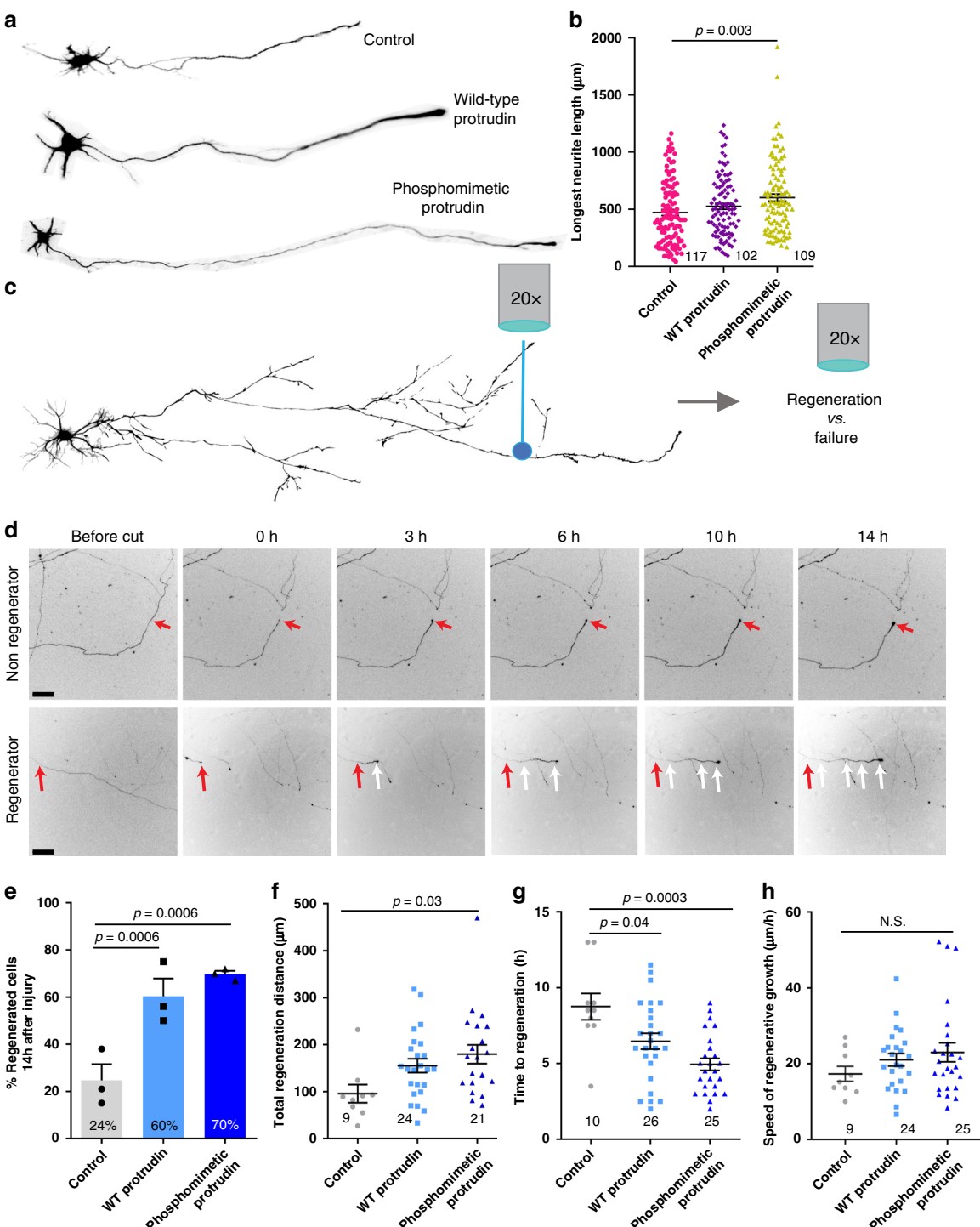

mutants in accordance with previous literature[21,25,27]. We deleted the Rab-binding domain (RBD) which is required for Rab11 anterograde transport[21]. We also created a mutant which lacked the FFAT (important for VAPA binding at ER contact sites) and the coiled-coil domain which has previously been shown to disrupt the interaction of Protrudin with the anterograde axonal motor KIF5[27]. We termed this mutant KIF5/VAPA mutant. In addition, we made a dominant-negative FYVE domain mutant which prevents the interaction of Protrudin with phosphoinositides on endosomal membranes[25] (Fig. 3e). Each of these mutants was separately expressed in cortical neurons at 10 DIV and their effects on the regeneration of mature axons were quantified at 13–17 DIV using

laser axotomy. Unexpectedly, overexpression of the RBD mutant caused extensive neuronal cell death (Supplementary Fig. 5c, d), indicating an essential role for the Protrudin-Rab11 interaction in neuronal viability but precluding examination of its effect on axon regeneration. Mutations of the KIF5/VAPA or the FYVE domains sharply diminished the effects of Protrudin overexpression on axon regeneration, whilst phosphomimetic Protrudin again stimulated robust regeneration (Fig. 3f). These data demonstrate that the interactions between Protrudin, endosomes and KIF5 are required for the axon regeneration-promoting effects of Protrudin, and that Protrudin-driven regeneration is accompanied by an increase in the anterograde transport of integrins and Rab11 endosomes.

**Fig. 2 Protrudin overexpression has a modest effect on initial neurite outgrowth but is a strong promoter of axon regeneration after laser axotomy.**
**a** Example neurons at 4 DIV overexpressing control construct, wild-type or phosphomimetic Protrudin. **b** The average length of the longest neurite in each condition ($n = 3$ independent experiments; 117 control, 102 WT and 109 phosphomimetic Protrudin cells were analysed, $p = 0.006$, Kruskal–Wallis with Dunn's multiple comparison test, Kruskal–Wallis statistic = 10.34). Error bars represent mean ± SEM. **c** Diagram of the laser axotomy method.
**d** Representative images show a regenerating and a non-regenerating axon over 14 h post laser axotomy. The red arrows at 0 h post injury shows the point of injury. The white arrows trace the path of a regenerating axon. Scale bars are 50 μm. **e** Percentage of regenerating axons overexpressing either mCherry control ($n = 3$ independent experiments, 45 neurons), mCherry wild-type Protrudin ($n = 3$ independent experiments, 45 neurons) or mCherry phosphomimetic Protrudin ($n = 3$ independent experiments, 39 neurons) (Fisher's exact test with analysis of stack of $p$ values and Bonferroni–Dunn multiple comparison test). Error bars represent mean ± SEM. **f** Quantification of regeneration distance 14 h after injury of control ($n = 9$ cells), WT ($n = 24$ cells) and phosphomimetic Protrudin ($n = 21$ cells) axons ($n = 3$ independent experiments, One-way ANOVA, $p = 0.04$, F statistic = 3.618). Error bars represent mean ± SEM. **g** Quantification of regeneration initiation time of control ($n = 10$ cells), WT ($n = 26$ cells) and phosphomimetic Protrudin ($n = 25$ cells) axons ($n = 3$ independent experiments, One-way ANOVA, $p = 0.0004$, F statistic = 9.076). Error bars represent mean ± SEM. **h** Quantification of the speed of regeneration of control ($n = 9$ cells), WT ($n = 24$ cells) and phosphomimetic Protrudin ($n = 25$ cells) axons ($n = 3$ independent experiments One-way ANOVA, $p = 0.348$, F statistic = 1.078). Error bars represent mean ± SEM.

**Protrudin promotes regeneration through interaction with the ER.** There is increasing evidence that endosomal transport is heavily influenced by ER-endosome contact sites, and that the distribution and morphology of ER tubules is controlled by kinesin-dependent endosomal transport[22,30]. In addition, ER-endosome and ER-plasma membrane contact sites have been observed in both axons and dendrites[31]. We reasoned that the localization of Protrudin to the ER, and its interaction with contact site proteins might contribute to its regenerative effects. In order to study this, we mutated the FFAT domain which is important for Protrudin's interaction with VAP proteins at ER contact sites[23], and we created a mutant lacking all three trans-membranes (TM1-3) domains which confer its membrane localization within the ER (Fig. 4a). The deletion of these hydrophobic regions releases Protrudin from the ER, rendering it cytosolic[23].

The ER exists as a continuous tubular organelle through axons (similar to an axon within the axon), and its genetic disruption causes axonal degeneration[32]. Re-establishment of the axonal ER may be equally as important as the re-establishment of the axon membrane for successful regeneration. Because ER tubules undergo highly dynamic movements, partly by hitchhiking on motile endosomes[33], we hypothesized that linkage of over-expressed Protrudin to kinesin might lead to an increase in tubular ER in the axon. To examine the effects of Protrudin overexpression on axonal ER we analyzed the distribution of endogenous reticulon 4 (RTN4) by immunofluorescence, which reports on ER abundance in axons[34] (Fig. 4b). Overexpression of wild-type and phosphomimetic Protrudin resulted in increased RTN4 in the growth cone shaft and at the axon tip of uninjured axons with a trend of having more ER in neurons expressing phosphomimetic Protrudin compared to those with wild-type Protrudin. Neurons expressing either the FFAT deletion mutant or the ER-membrane deletion mutant of Protrudin (lacking TM1-3 domains) did not exhibit an accumulation of RTN4 in the distal axon (Fig. 4b–d). We observed a similar accumulation of ER at the tip of protrusions in HeLa cells expressing phosphomimetic but not wild-type Protrudin (Supplementary Fig. 1d). To confirm our neuronal findings, we studied the distribution of an additional smooth ER marker—REEP5 (Supplementary Fig. 6). We found an accumulation of overexpressed REEP5 at the growth cone in neurons expressing wild-type and phosphomimetic Protrudin with phosphomimetic Protrudin having the most robust effects (Supplementary Fig. 6). These findings indicate that phosphomimetic, active Protrudin has stronger effects on ER mobilization compared with wild-type Protrudin, in contrast to its effects on Rab11 transport (Fig. 3).

In order to study the importance of the Protrudin-ER interaction for Protrudin-mediated axon regeneration, each of the mutants described above was separately expressed in primary rat cortical neurons at 10 DIV and their effects on axon regeneration were studied at 13–17 DIV using laser axotomy. Both mutants sharply diminished the effects of Protrudin overexpression on axon regeneration compared to phosphomimetic Protrudin which stimulated robust regeneration (Fig. 4e). However, both deletion mutants had moderate regenerative effects compared to control neurons. Whilst these were not statistically significant, they indicate that Protrudin may exert some effects independently of its localization to ER contact sites.

We showed above that a combined KIF5/VAPA mutant which disrupts binding to KIF5 reduced the regenerative effect of phosphomimetic Protrudin (Fig. 3f). Interestingly, disruption of the FFAT domain alone had a similar effect on suppressing axon regeneration to the combined KIF5/VAPA deletion mutant, underlying the importance of Protrudin's interaction with ER contact site protein VAPA to mediate axon regeneration. These data demonstrate that Protrudin enables the enrichment of ER in axon growth cones, and that this supports Protrudin's regenerative effects. This indicates an important role for the ER in mediating CNS axon regeneration.

Collectively, the results so far demonstrate that Protrudin enables axon regeneration by acting as a scaffold that links key players that participate in regeneration. Axonal ER, recycling endosomes, kinesin 1, and phosphoinositides, are all brought together in distal axons and regenerating growth cones. The finding that mutation of any of the binding domains in Protrudin abrogates its effect on regeneration suggests that the co-location of all these components is necessary for efficient axon regeneration.

**Overexpression of Protrudin promotes neuronal survival in the retina and axon regeneration in the injured optic nerve.** Protrudin's robust effect on CNS axon regeneration in vitro prompted us to investigate its effectiveness on optic nerve regeneration. We first examined Protrudin mRNA levels in RGCs in published RNA-sequencing datasets[35] and found that Protrudin mRNA is present at low levels in mature, adult RGCs (Fig. 5a). This corresponded with our findings in cortical neurons but not in regenerative PNS neurons where Protrudin levels are much higher (Fig. 1c, d). We generated three constructs for AAV delivery to the retina by intravitreal injection: AAV2-GFP, AAV2-ProtrudinGFP, and AAV2-phosphomimetic-Protrudin-GFP. The viruses transduced 40–45% of RGCs throughout the retina and the protein was observed throughout uninjured axons (Supplementary Fig. 7a, b). Higher Protrudin levels were detected by immunohistochemistry of wholemount retinas in eyes following overexpression of wild-type or phosphomimetic Protrudin compared to the control virus (Supplementary Fig. 7c). Viruses were injected 2 weeks before optic nerve crush, and 2 weeks after

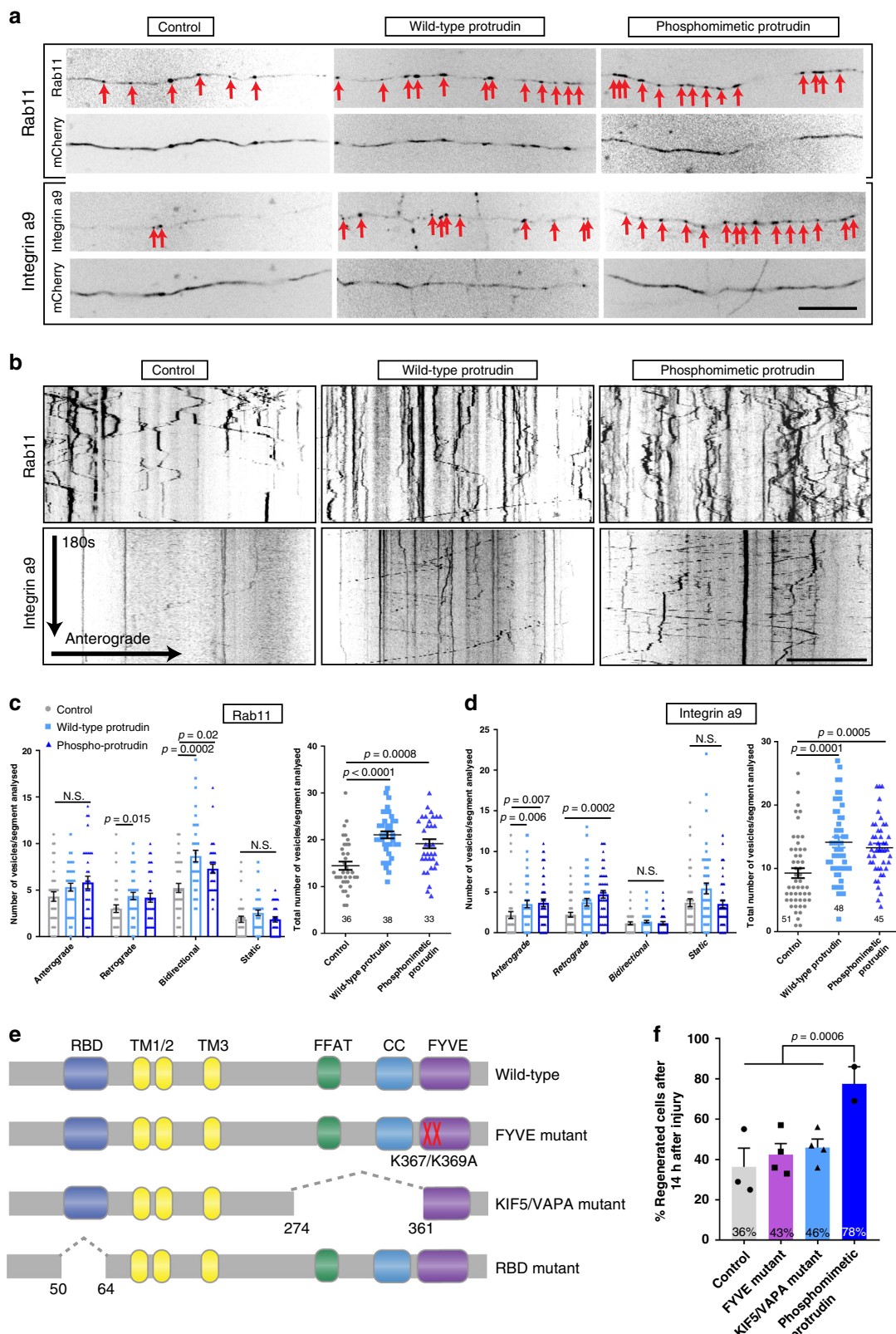

injury the anterograde axon tracer—cholera toxin subunit-ß (CTB) was administered (Fig. 5b). We measured RGC survival in the retina and axon regeneration in the optic nerve. At 2 weeks after the injury we quantified RGC survival by counting the number of RBPMS-positive RGCs in wholemount retinas. We found that retinas expressing phosphomimetic Protrudin had a

significantly higher percentage of surviving RGCs (52%) compared to those injected with GFP control (28%) whilst transduction with wild-type Protrudin (27%) had no effect (Fig. 5c, d). In contrast, both wild-type and phosphomimetic Protrudin had robust effects on axon regeneration. Optic nerves of mice expressing GFP exhibited limited regeneration (0% >0.5 mm from

**Fig. 3 Protrudin enhances the transport of growth machinery and receptors in the distal axon, and its involvement in axon transport is required for axon regeneration. a** Representative distal axon sections of neurons expressing integrin α9-GFP or Rab11-GFP, together with either mCherry (control), mCherry-wild-type Protrudin or mCherry-phosphomimetic Protrudin. Scale bar is 20 µm. **b** Kymographs showing the dynamics of integrin α9-GFP and Rab11-GFP in distal axons of co-transfected neurons. Scale bar is 10 µm. **c** Quantification of Rab11-GFP axon vesicle dynamics and total number of Rab11 GFP vesicles in distal axon sections ($n = 3$ independent experiments) (for transport, Kruskal–Wallis with Dunn's multiple comparison test was used; anterograde—$p = 0.175$, Kruskal–Wallis statistic = 3.489; retrograde—$p = 0.02$, Kruskal–Wallis statistic = 8.197, bidirectional—$p = 0.0002$, Kruskal–Wallis statistic = 16.60, static—$p = 0.105$, Kruskal–Wallis statistic = 4.499; for total number of vesicles, one-way ANOVA was used—$p < 0.0001$, F statistic = 15.68). Error bars represent mean ± SEM. **d** Quantification of integrin α9-GFP axon vesicle dynamics and total number of integrin α9-GFP vesicles in distal axon sections ($n = 3$ independent experiments) (for transport, Kruskal–Wallis with Dunn's multiple comparison test was used; anterograde—$p = 0.002$, Kruskal–Wallis statistic = 12.57; retrograde—$p = 0.0003$, Kruskal–Wallis statistic = 16.64, bidirectional—$p = 0.271$, Kruskal–Wallis statistic = 2.610, static—$p = 0.051$, Kruskal–Wallis statistic = 5.951; for total number of vesicles—$p < 0.0001$, Kruskal–Wallis statistic = 21.81). Error bars represent mean ± SEM. **e** Schematic representation of Protrudin transport domain mutants. **f** Percentage of regenerating axons in neurons expressing mCherry-Protrudin domain mutants—FYVE ($n = 4$ independent experiments, 56 neurons) and KIF5/VAPA ($n = 4$ independent experiments, 56 neurons) compared to phospho-Protrudin as a positive control ($n = 2$ independent experiments, 24 neurons), and mCherry as a negative control ($n = 3$ independent experiments, 42 neurons) (Fisher's exact test with analysis of stack of $p$ values and Bonferroni–Dunn multiple comparison test). Error bars represent mean ± SEM.

the crush site), while regenerating axons extended up to 2.75 mm in wild-type Protrudin-transduced animals, and as far as 3.5 mm in phosphomimetic Protrudin-transduced animals. The numbers of regenerating axons were high, particularly for phosphomimetic Protrudin, in which over 630 axons were seen proximally, significantly more than in control (44 axons) or in wild-type Protrudin (380 axons) (Fig. 5e, f). Co-localization between CTB and GAP43 was found throughout the nerve in all conditions suggesting that the majority of CTB-positive axons observed in the nerve past the injury site are regenerating axons (Supplementary Fig. 7d). These findings confirm that phosphomimetic Protrudin expression leads to a substantial increase in neuronal survival and axon regeneration in the injured CNS, whilst wild-type Protrudin overexpression does not protect RGCs from death, but has robust effects on the regeneration of surviving neurons.

**Overexpression of wild-type or phosphomimetic Protrudin is neuroprotective in vivo.** We next examined the effects of Protrudin expression using an additional RGC neuroprotection model. Because optic nerve injury leads to severe neuronal loss two weeks after injury (typically 80–90%) we used an acute retinal explant model which is often used to detect potential neuroprotective treatments for glaucoma[36,37]. Viruses were injected intravitreally, and retinas were removed two weeks later and cultured as explants for three days (Fig. 6a). Both wild-type and phosphomimetic Protrudin were entirely neuroprotective, with these retinas exhibiting no loss of RGC neurons, whilst GFP-only controls lost 55% of their RGCs (Fig. 6b, c). In addition, both Protrudin constructs showed widespread general neuroprotection as there was no reduction in the DAPI-positive cells after injury (Fig. 6d).

## Discussion

Our study demonstrates axon regeneration in the CNS driven by overexpression of the scaffold molecule, Protrudin. Protrudin enables robust axon regeneration and neuroprotection in the retina and optic nerve and promotes regeneration after axotomy of cortical neurons in vitro. The action of Protrudin is to bind recycling endosomes, ER and kinesin and carry them and their contents to the tip of axons and growth cones.

Protrudin mRNA is expressed at low levels in cultured CNS cortical neurons compared to other abundantly expressed proteins (Fig. 1c). In previous reports, Protrudin has been detected in mouse primary hippocampal neurons at 1 DIV predominantly localized to the pericentrosomal compartment and to growing

neurites, and present in dendrites and at the growth cone[21]. Here, we examined the distribution of the endogenous protrudin protein in rat primary cortical cultures throughout development[19]. We found that Protrudin's distribution changes with neuronal maturity. Protrudin is present in the newly extending axons but as neurons mature and polarize, the Protrudin protein is redistributed towards the cell body and dendrites, suggesting that the protein is not available to participate in regeneration (Fig. 1e–g). Many molecules become selectively excluded from axons and directed to dendrites as neurons mature and compartmentalize. Previously we have studied the selective distribution of Rab11 and integrins which are essential for neurite outgrowth and axon transport[17,19]. For these molecules the exclusion from axons is more complete than for Protrudin which is driven in axons when the levels in the cell body are raised by overexpression. The exclusion of growth-related molecules from mature axons is one of the reasons for their failure to regenerate, and restoration of Rab11 vesicles and integrins to mature axons can restore regeneration[17,19,38].

The main aims of this study were to determine whether Protrudin can enable axon growth and regeneration and investigate its mechanism of action. Previously, the ability of Protrudin to stimulate process outgrowth was shown to be dependent on its phosphorylation. Because receptors able to activate proteins by phosphorylation may be sparse in mature CNS axons[39], a construct for phosphomimetic, constitutively active Protrudin was made, based on the previously identified phosphorylation sites[21].

Previous reports have shown that overexpression of Protrudin can enhance neurite outgrowth in PC12 cells and in hippocampal neurons at early stages of development[21]. Overexpression of Protrudin during early neurite extension upon dissociation led to a modest increase in axon length when phosphomimetic Protrudin was overexpressed in rat cortical neurons (Fig. 2a, b). However, we did not observe any differences when wild-type Protrudin was overexpressed, as predicted by previous studies in hippocampal neurons[21].

To measure the effects of Protrudin on axon regeneration we used a culture model in which neurons mature over time, and the probability of axons regenerating declines from around 70 to 5% by 20 DIV[19]. These in vitro regeneration experiments demonstrate that wild-type and phosphomimetic Protrudin greatly enhance axon regeneration after laser injury (Fig. 2c–h). The percentage of regenerating axons especially in the phosphomimetic Protrudin condition (70%) was found to be higher than some of the best treatments utilised previously in this model system such as depletion of EFA6—an ARF6 activator in the axon (59%)[18] and

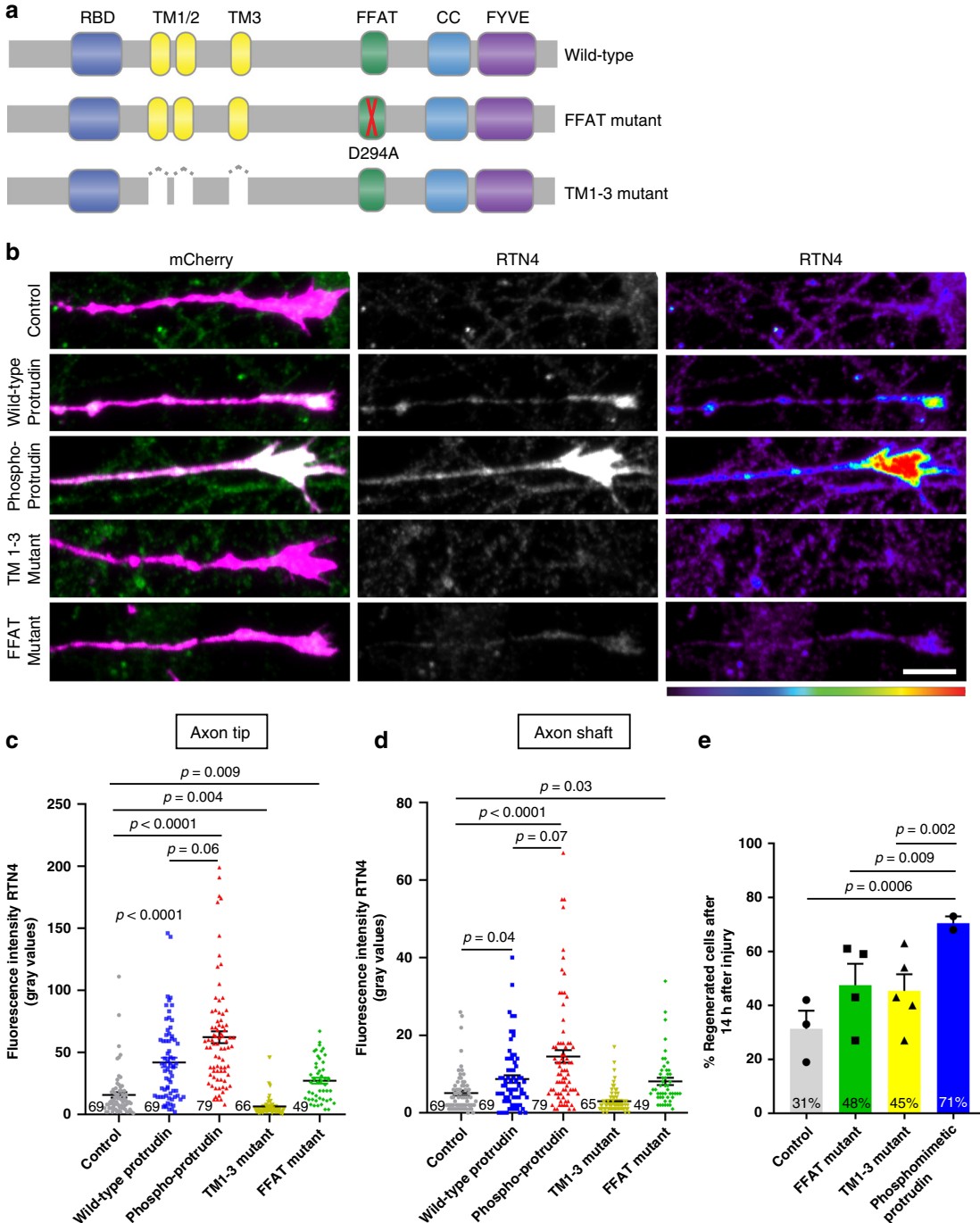

**Fig. 4 Protrudin overexpression enhances ER presence at growth cones and this interaction is required for successful axon regeneration. a** Schematic representation of Protrudin endoplasmic reticulum domain mutants. **b** Representative images of RTN4 immunofluorescence (green) in the distal axon of neurons expressing the indicated m-Cherry Protrudin constructs (magenta). Scale bar is 10 μm. **c, d** Quantification of RTN4 fluorescence intensity at the axon tip and shaft ($p < 0.0001$, Kruskal–Wallis with Dunn's multiple comparisons test). Error bars represent mean ± SEM. **e** Percentage of regenerating axons in neurons expressing mCherry-Protrudin domain mutants—FFAT ($n = 4$ independent experiments, 60 neurons) and TM1-3 ($n = 5$ independent experiments, 45 neurons) compared to phosphomimetic Protrudin as a positive control ($n = 2$ independent experiments, 21 neurons), and mCherry as a negative control ($n = 3$ independent experiments, 41 neurons). Error bars represent mean ± SEM.

overexpression of dominant-negative Rab11 (38%)[19]. Seventy percent appears to be the ceiling value for regeneration in this assay. Overexpression of wild-type Protrudin was also capable of enhancing axon regeneration but to a lesser extent than phosphomimetic Protrudin.

Once Protrudin's regenerative ability was confirmed, we studied potential mechanisms, based on Protrudin's many interaction domains. Protrudin's actions affected transport as predicted, because overexpression of wild-type and phosphorylated Protrudin resulted in increased transport of Rab11 endosomes and in enhanced antero-grade and retrograde transport of integrins in the distal axon (Fig. 3a–d). This study focused on integrin α9 because of its ability to promote long-range axon regeneration in the spinal cord and on Rab11 because these endosomes transport integrins[15,17]; however,

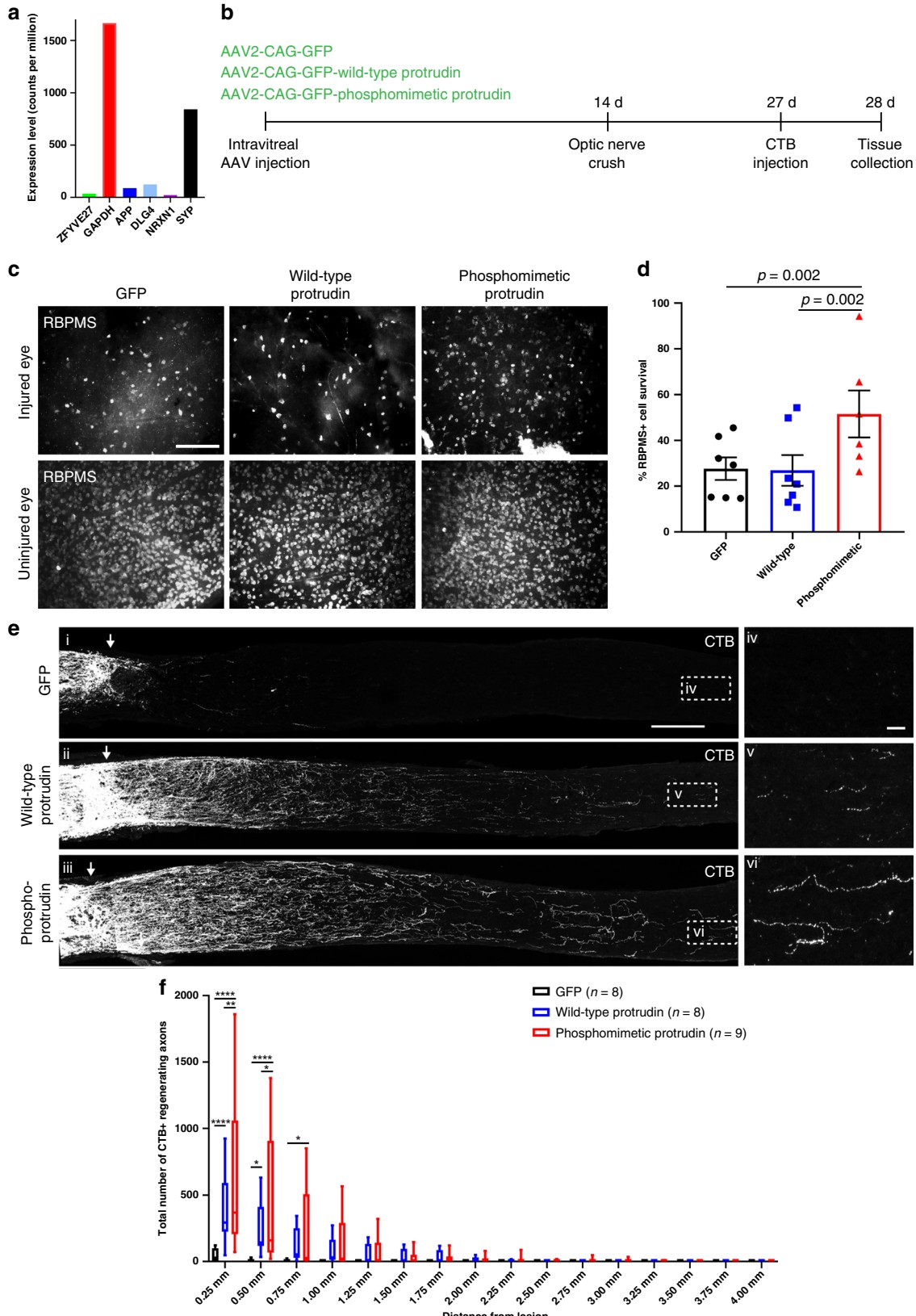

the effects of Protrudin on axon growth are most likely both integrin dependent and independent[29]. In addition to integrins, Rab11 endosomes transport many growth-promoting molecules that could influence regeneration including the IGF-1 and TrkB receptors[40–42]. Rab11 may also promote regeneration through regulation of membrane trafficking events[43]. In addition, based on the original hypothesis, the domain's linking axonal transport to membrane and endosomal trafficking (FYVE domain, KIF5/VAPA domain and RBD) were mutated, and each of these changes suppressed the axon regeneration-promoting effect of Protrudin (Fig. 3e, f).

Fig. 5 Protrudin enhances regeneration of RGC axons following optic nerve crush. a Protrudin mRNA levels during the progression of glaucoma in comparison to other neuronal markers. b Experimental timeline for optic nerve crush. c Representative images of retinal wholemounts stained for RBPMS (white) to label retinal ganglion cells in the uninjured and injured for each condition 2 weeks after optic nerve crush. Scale bars are 100 μm. d Quantification of RGC survival 2 weeks post crush. Eyes injected with phosphomimetic Protrudin have a higher percentage of RGC survival ($n = 5$ animals) compared to control ($n = 7$ animals, $p = 0.007$) or wild-type Protrudin ($n = 7$ animals, $p = 0.002$) (Fisher's exact test with analysis of stack of p values and Bonferroni–Dunn multiple comparison test). Error bars represent mean ± SEM. e CTB-labelled axons in the optic nerves of mice transduced with viruses for wild-type Protrudin, phosphomimetic Protrudin and GFP control. Arrows indicate lesion site. Insets (iv-vi) show regenerating axons in the distal optic nerve. Scale bar is 200 μm and on inset is 20 μm ($n = 8$–9 animals/group). f Quantification of regenerating axons at increasing distances distal to the lesion site, displayed as mean ± SEM. Statistical significance was determined by two-way ANOVA with Bonferroni post hoc test for multiple comparisons. $**p < 0.005$, $***p < 0.001$, $****p < 0.0001$. Individual p values are as follows: $p < 0.0001$ for GFP vs. WT and GFP vs. phosphomimetic Protrudin at 0.25 mm, $p = 0.003$ for WT vs. phosphomimetic Protrudin at 0.25 mm, $p = 0.01$ for GFP vs. WT at 0.5 mm, $p < 0.001$ for GFP vs. phosphomimetic Protrudin at 0.5 mm, $p = 0.02$ for WT vs. phosphomimetic Protrudin at 0.5 mm and $p = 0.01$ for GFP vs. phosphomimetic Protrudin at 0.75 mm. The box plots show the first and third quartiles (the box limits), the median (horizontal line), and the minimum and maximum values (whiskers).

The involvement of the ER in axon regeneration has not been extensively studied in mammalian CNS neurons before. One previous study has presented correlative evidence implicating not only the ER but also the protrudin-binding protein—spastin in axon but not dendrite regeneration due to their accumulation at the tips of regenerating axons after axonal injury in *Drosophila*[44] and ER-microtubule interaction is involved in the establishment of neuronal polarity[34]. Here, we found that phosphomimetic Protrudin overexpression increased ER (shown by RTN4 and REEP5, markers of smooth ER)[34] in axonal growth-cones (Fig. 4b–d). In addition, TM1-3 deletion mutant and FFAT deletion mutant Protrudin, both important for ER localization, each eliminated its ability to enrich ER at growth cones to the same extent as wild-type or phosphomimetic Protrudin and to promote axon regeneration after laser axotomy (Fig. 4). There are several mechanisms by which enrichment of the ER could facilitate axon regeneration, including bulk transfer of lipids from the ER to the plasma membrane, synthesis and transfer of signalling lipids, calcium signalling and involvement in organelle trafficking. ER-plasma membrane interaction sites are a potential site for some of these interactions[22,45]. Protrudin-mediated enrichment of ER into the tip of growing processes could function to enable the transfer of lipids, with the FYVE domain promoting interaction with the surface membrane[25], allowing for rapid expansion of the growth cone plasma membrane—a requirement for successful axon regeneration[46].

The overall result of these studies into the interaction domains of Protrudin is that all of them are involved in axon regeneration, and that inactivation of any of them removes most of the ability of Protrudin to promote regeneration. The conclusion is that Protrudin works by bringing all of its binding partners together in such a way that they can collaborate in enabling axon regeneration. An outstanding issue is the relative contribution of the ER and Rab11 to the regenerative effects of Protrudin and phosphomimetic Protrudin, especially because the ER is closely linked to numerous types of endosomes through interactions at contact sites. Further work is needed to determine if additional interventions which increase axonal ER also lead to enhanced regeneration, independently of a direct interaction with Rab11.

The next step was to examine Protrudin's effects in vivo. The optic nerve crush model has proven an excellent screen for regeneration treatments. Promoting RGC regeneration has the potential to restore vision loss associated with optic neuropathies such as glaucoma, and virally delivered gene therapy for eye disease is already in clinical practice[47]. In the current study, adult mice were treated by delivering an AAV vector into the vitreous 2 weeks before an optic nerve crush. Both phosphomimetic and wild-type Protrudin led to a large number of axons regenerating for a long distance only 2 weeks after optic nerve crush with only phosphomimetic Protrudin having a pronounced effect on neuronal survival 2 weeks post crush (Fig. 5). Expression of phosphomimetic Protrudin allowed for 400–500 neuronal fibres to reach the 0.5 mm mark by 2 weeks after injury suggesting that this intervention is comparable to the most potent interventions reported to date.

In addition, Protrudin's overexpression (both wild-type and phosphomimetic forms) was completely neuroprotective in a retinal explant model of RGC injury and 2 weeks post optic nerve crush (Figs. 5, 6). This effect could be due to increased signalling as a result of improved axonal transport which in turn activates retrograde survival signals. Further studies are needed in order to pinpoint the exact mechanism of Protrudin-driven neuroprotection in the eye.

Our study demonstrates robust axon regeneration in the adult CNS driven by overexpression of the adapter protein Protrudin. Overexpression of Protrudin, particularly in its phosphomimetic, active form greatly enhanced regeneration in cortical neurons in vitro and in the injured adult optic nerve. Importantly, both wild-type Protrudin and phosphomimetic Protrudin expression lead to an accumulation of ER and enhanced axonal transport in the distal axon and interfering with Protrudin's ER localization or transport domains abrogates its regenerative effects, indicating a central role for these processes in mediating Protrudin-driven regeneration. We propose that Protrudin enables regeneration by acting as a scaffold to link the ER, recycling endosomes, kinesin-based transport and membrane phospholipids. Our findings establish the importance of these components in facilitating CNS axon regeneration, whilst suggesting Protrudin gene-therapy as a potential approach for repairing CNS axon damage.

## Methods

**DNA constructs**. Human Protrudin constructs (in pmCherry-C1 and pEGFP-C1)[48], CMV-integrin-alpha9-GFP[29] and CMV-Rab11 -GFP[19,29] were used. The CMV promoter in all constructs was replaced by a human synapsin (Syn) promoter by Gibson assembly cloning. The viral vector plasmid backbones (AAV2-sCAG-GFP) were a kind donation by Prof. Joost Verhaagen, The Netherlands Institute for Neuroscience. Protrudin-GFP was cloned from pEGFP-C1 plasmid into viral vector plasmids using Gibson cloning. Site-direct mutagenesis was performed in order to create the Protrudin active phosphomimetic form (QuikChange II Site-Directed Mutagenesis Kit, Agilent Technologies). All primers used for Protrudin cloning are described in Table S1. All constructs were verified by DNA sequencing.

**Cell culture and transfections**. Rat cortical neurons were dissected from E18 embryos from Sprague Dawley rats and plated on imaging dishes or on acid-washed glass coverslips at the following densities: $1 \times 10^5$ cells/dish for immuno-cytochemistry, $2 \times 10^5$ for axotomy or live-cell imaging and $8 \times 10^4$ cells/coverslip. All surfaces were coated with poly-D-lysine (Sigma, D1149-100MG), diluted in borate buffer to a final concentration 50 μg/mL. The cells were grown in serum-free MACS Neurobasal Media supplemented with 2% NeuroBrew21 and 1% GlutaMAX supplements at 37 °C in 7% $CO_2$ incubator. Cortical neurons were transfected using NeuroMag magnetofection (Oz Biosciences, NM50200) system where 7 μg of DNA plasmid is mixed with 100 μL NB media and 8 μL of magnetic beads. The reaction was kept for 30 min at 37 °C before adding 900 μL of pre-warmed NB media to a final volume of 1 mL. The original neuronal media was removed, and 1 mL of

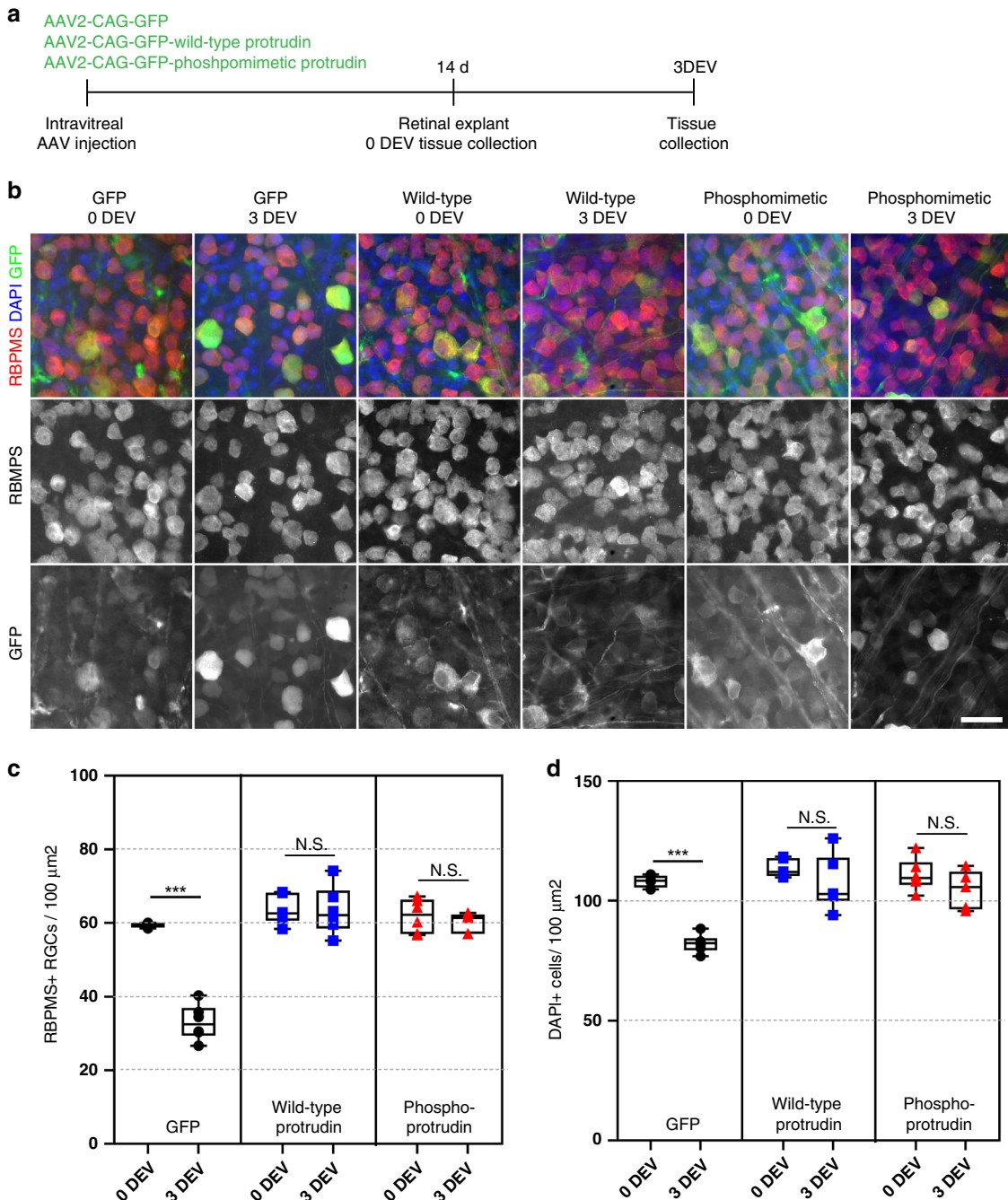

**Fig. 6 Protrudin is neuroprotective to RGCs and other cell types in the retina after a retinal explant. a** Experimental timeline for retinal explant experiment. **b** Representative images of RGCs (red for RBPMS) 0 and 3 days ex vivo (DEV) in eyes injected with control virus, wild-type Protrudin or phosphomimetic Protrudin (green for GFP) and stained for DAPI (blue). Scale bar is 20 μm. **c** Quantification of RGC survival in retinal explant ($n = 4$–6 animals for each condition) (Two-tailed Student's $t$-test). **$p < 0.005$, ***$p < 0.001$, ****$p < 0.0001$. Individual p values are as follows: $p < 0.0001$ for GFP, $p = 0.945$ for WT and $p = 0.439$ for phosphomimetic Protrudin when compared at 0 DEV to 3 DEV. The box plots show the first and third quartiles (the box limits), the median (horizontal line), and the minimum and maximum values (whiskers). The circles, squares and rectangles represent individual data points. **d** Quantification of DAPI-positive cell survival in retinal explant ($n = 4$–6 animals for each condition) (Two-tailed Student's $t$-test). **$p < 0.005$, ***$p < 0.001$, ****$p < 0.0001$. Individual $p$ values are as follows: $p < 0.0001$ for GFP, $p = 0.234$ for WT and $p = 0.189$ for phosphomimetic Protrudin when compared at 0 DEV to 3 DEV. The box plots show the first and third quartiles (the box limits), the median (horizontal line), and the minimum and maximum values (whiskers). The circles, squares and rectangles represent individual data points.

transfection mixture was added. Dishes were then incubated at 37 °C for 30 min on a magnetic plate before the original media was returned on the plates. Plasmid reporter gene expression was observed 48 h post-transfection.

**Antibodies**. Rabbit anti-Zfyve27 (Proteintech, 12680-1-AP, 1:500), Mouse Pan-Neurofascin (extracellular) (NeuroMab, 75–172, 1:50), Rabbit anti-Reticulon 4

(Novus Biologicals, NB100-56681, 1:250), Rabbit anti-RBPMS (Phosphosolutions, 1830-RBPMS, 1:500), Mouse anti-mCherry (ClonTech, 632543, 1:500), DAPI (ThermoFisher Scientific, D3751, 1:10,000), Sheep anti-GAP43 (kind donation from the Benowitz lab, 1:5000), mouse monoclonal anti-beta actin (C4) HRP conjugated (Santa Cruz, sc-47778, 1:5000), goat anti-rabbit IgG HRP conjugated (Sigma, A4914, 1:80000),Goat anti-rabbit 488 (ThermoFisher Scientific, A27034, 1:500), Goat anti-rabbit 568 (ThermoFisher Scientific, A-11011, 1:500), Goat anti-

rabbit 647 (ThermoFisher Scientific, A27040, 1:500), Goat anti-mouse 647 (ThermoFisher Scientific, A-21235, 1:500), Goat anti-mouse 568 (ThermoFisher Scientific, A-11031, 1:500).

**Immunostaining.** Cortical neurons were fixed in 3% PFA for 15 min and then thoroughly washed and kept in PBS at 4 °C. Cells were permeabilized in 3% BSA in PBS and 0.1% Triton for 5 min and then blocking solution was added (3% BSA in PBS) for 1 h at room temperature. Primary antibodies were added at the correct concentration and kept for 1.5 h at room temperature. Antibodies were then washed three times in PBS for 5 min. Secondary antibodies were applied at the correct concentration for 1 h at room temperature in a dark chamber. The cells were then washed three times in 1xPBS and mounted using coverslips and Diamond anti-fade mounting agent with DAPI (Molecular Probes) or FluorSave mounting reagent (Calbiochem). Mice were anesthetized using 1–2% isoflurane and transcardially perfused with PBS followed by 4% paraformaldehyde (PFA). Optic nerves were dissected and immersed in 4% PFA. The tissue was post-fixed overnight, then immersed in 30% sucrose for 24 h for cryoprotection. Tissue was embedded in Tissue-Tek OCT and snap-frozen for cryosectioning. 14-μm-thick longitudinal sections of the optic nerve were obtained on charged Superfrost microscope slides using a Leica CM3050 cryostat. Slides were dried and stored at −20 °C.

**Confocal microscopy.** Images of immunostained cells were taken with a confocal microscope (Leica DMI 4000 B) using LAS-AF software (Leica, Version 2.7.3.9723). For Protrudin localisation, a z-stack of images was obtained through each cell by taking an image at every 0.5-μm thickness and an average intensity z-projection was created in Fiji software[49]. All images in an experiment were taken using the same microscope settings. The intensity of processes was measured by placing a line shape (usually about 50 μm) on processes and taking a background intensity of the same shape immediately next to each process. The background was then subtracted from the intensity at each process. For Protrudin intensity measurements in GFP-transfected neurons, regions of interest were selected in the 488 channel and a line was placed on each process and immediately next to it to measure background. This strategy immediately excludes any biases on selecting processes with higher intensity of protrudin staining. For all experiments measuring axon-to-dendrite ratio, the intensity in dendrites was an average from measurements in at least 3 dendrites per cell where possible.

**Live-cell imaging.** Live-cell imaging was performed using spinning disk confocal microscopy, using an Olympus IX70 microscope with a Hamamatsu EM-CCD Image-EM camera and a PerkinElmer Ultra-VIEW scanner. Videos were taken using Meta-Morph software (Version 7.6.1.0). Rab11 and integrin vesicle trafficking along the axon was imaged at the proximal (up to 100 μm) and distal part (beyond 600 μm) of axons of neurons transfected with Protrudin (as described before in[18,29]). In short, one image per second was obtained for 3 min. Kymographs were obtained by measuring an individual axon segment. Anterograde, retrograde, bidirectional and static modes of transport were measured. The percentage of co-localization between integrin or rab11 and Protrudin was calculated as the number of vesicles containing either was divided by the total number of vesicles. All analysis was performed using Meta-Morph software (Version 7.6.1.0).

**Western blotting.** PC12 cells were transfected using lipofectamine. Forty-eight hours later cells were lysed using the cOmplete Lysis Kit (Roche). Cells were washed with ice-cold PBS. Five-hundred microliters of pre-cooled lysis buffer was added to each well of a 6-well plate and the lysate was scraped using a cell scraper and transferred to a sterile 1.5 mL Eppendorf. The lysate was incubated on ice for 30 min with occasional mixing. The samples were then centrifuged at 10,000 × g at 4 °C for 10 min. The supernatant was transferred to 1.5 mL Eppendorf and the pellet was discarded. The total protein concentration was measure by BCA assay using Pierce BCA Assay Kit Protocol (ThermoFischer Scientific). The 96-well plate containing the sample lysates and BCA reagents was analyzed using Gen5.1 software and concentrations were derived from a standard curve for albumin control. 15 μg of PC12 cell lysate was then treated with LDS Sample Buffer NuPAGE 4x (1:4, ThermoFisher Scientific) and Sample Reducing Agent (1:10, ThermoFisher Scientific) and were analyzed by Western blot. Samples were run on a 4–12% gel at 120 V at room temperature in 40 mL Running Buffer NuPAGE (ThermoFisher Scientific) diluted in H20 to 800 mL. The gel was then transferred onto a nitrocellulose membrane (Invitrolon PDVF/Filter Paper Sandwich, ThermoFischer Scientific) for 1.5 h at 40 V at 4 °C in 50 mL Transfer Buffer NuPAGE (ThermoFisher Scientific) in 100 mL methanol, topped up with ddH20 to 1 L. The membrane was then blocked either in 5% milk or 3% BSA depending on the antibody for 1 h and incubated overnight with the primary antibody diluted in the blocking solution to the right concentration at 4 °C. The membrane was then rinsed three times in Tris-buffered saline with Tween 20 (TBST buffer) for 10 min each. The TBST buffer was removed. Secondary peroxidase-conjugated antibodies were diluted to the right concentration in blocking solution and were then added for 1 h at room temperature. SuperSignal West Dura Extended Duration Substrate kit (ThermoFischer Scientific) and Alliance software (Version 16.05) were then used for detection.

**Axotomy.** 10 DIV neurons were transfected with various constructs using magnetofection as described above. Between 13–17 DIV, their regeneration ability was examined using the laser axotomy model described in detail in[19]. In short, axotomy was performed by an UV Laser (355 nm, DPSL-355/14, Rapp OptoElectronic, Germany) connected to a Leica DMI6000B (Leica Systems, Germany), and all images were taken with an EMCCD camera (C9100-02, Hamamatsu). Axons were cut at least 600 μm away from the cell body and regeneration was observed for 14 h post injury at 30-min intervals. If more than 50% neuronal cell death occurred in the axotomised cells, the experiment was excluded from the final analysis.

**Animal studies.** All procedures were performed in accordance with protocols approved by the Institutional Animal Care and Use Committee (IACUC) at the National Institutes of Health, by the UK Home Office regulations for the care and use of laboratory animals under the UK Animals (Scientific Procedures) Act (1986) and in accordance with the Swedish Board of Agriculture guidelines and were approved by the Karolinska Institutet Animal Care Committee. Pregnant female Sprague Dawley rats (Charles River, 8–12 weeks old) were housed in pathogen-free facility with free access to food and a standard 12-h light/dark cycle and embryos of both sexes were dissected at E18 for cortical cultures. Female C57Bl/6 mice aged 6–8 weeks (Charles River) were housed in a pathogen-free facility with free access to food and a standard 12-h light/dark cycle. All animals were housed at 18–23 °C and in 40–60% humidity. Intravitreal injections of viruses were administered 14 days prior to optic nerve crush or whole retinal explant. Two microliters of the injecting solution for mice was drawn into a sterile 5 μL Hamilton syringe (#65RN; Needle: ga33, 8 mm, pst2, Hamilton Co.). Attention was paid to avoid lens penetration, extraocular muscles and vortex vein impingement. The Hamilton syringe was then held in situ for 30 s before a sterile 30-gauge needle (B. Braun Medical Ltd.) was used to puncture the central cornea, reducing intraocular pressure and injection solution reflux, at which point the Hamilton syringe was carefully withdrawn. Separate needles were allocated to each virus to prevent contamination, and syringes were rinsed between injections with ethanol followed by sterile PBS.

**Optic nerve crush.** Optic nerve crush was performed as described previously[50] and as follows: Micro-scissors were used to make an incision in the conjunctiva and expose the optic nerve. Curved forceps were then inserted below the external ocular muscle, avoiding the ophthalmic artery and retrobulbar sinus, and positioned around the exposed nerve. The nerve was crushed for 10 s ~1 mm posterior to the eye. Eyes were then observed fundoscopically for signs of ischaemia, and mice were monitored for signs of intraorbital bleeding. Mice were given a subcutaneous injection of 1 mg/kg buprenorphine as an analgesic and topical application of ophthalmic ointment to prevent corneal drying. Intravitreal injections of CTB (1.0 μg/μL, Sigma) were administered 2 days prior to perfusion. 2 μL of the solution injected as described above. Images of optic nerves and retinas were obtained using the ZEN Digital Imaging Suite.

**Retinal wholemounts preparations.** Eyes were sharp dissected out of the orbit with Vanna scissors and immediately immersed in 4% PFA for 2 h before transfer to PBS. Retinal wholemounts were then prepared under a dissection microscope, flat mounted onto Milipore filter paper in 24-well plates and stored at 4 °C. Following three 10-min washes with PBS, TBST blocking buffer (3% goat serum, 1% BSA, 0.3% TritonX-100) was applied for 1 h at room temperature with gentle rotation. This was replaced with RBPMS primary antibody (Phosphosolutions, Aurora, USA; 1:500) in TBST blocking buffer at 4 °C overnight with gentle rotation. Following three further 10-min washes with PBS, Alexa Fluor 647 secondary antibody 1:500 in TBST blocking buffer at 4 °C overnight with gentle rotation. After three further washes with PBS, mounted onto charged Superfrost microslides using Fluorsave mounting reagent (Calbiochem) and allowed to dry for 4 h before imaging. Blinded manual counting of all images was undertaken (8 images/wholemount, 2 images/quadrant). Uninjured right eyes were counted (n = 3–4 per group) and the averaged used to normalize percentage RGC survival.

**Whole retina explant culture.** Mice were euthanized by cervical dislocation, eyes enucleated, and placed immediately into ice-cold HBSS. Retinas were dissected from the eyes in HBSS on ice, flat-mounted with the ganglion cell layer up on a cell culture insert (Millipore), and submerged in tissue culture media containing Neurobasal −A, 1% penicillin-streptomycin (10,000 U/ml), 1% glutamine (100×), 1% N-2 (100×) and 1% B-27 (50×) (all ThermoFisher Scientific). Retinas were incubated in 6-well plates at 37 °C and 4% $CO_2$ for 3 days and were fed by replacing 50% of the media on day 2. Retina were fixed in 3.7% PFA and stained with antibodies against RBMPS and GFP followed by counterstaining with DAPI. For untreated "DEV 0" controls, retinas were dissected and placed straight into 3.7% PFA.

**Statistics and reproducibility.** Statistical analysis was performed using GraphPad Prism 8.0 (GraphPad Software, La Jolla, CA). Each data set was individually tested for normal distribution using the D'Agostino–Pearson normality test. When data were normally distributed one-way ANOVA with multiple comparisons was used to test statistical significance between the experimental groups with Tukey's post hoc test. Several data sets were shown to be non-normally distributed. Therefore, a

non-parametric Kruskal–Wallis test with multiple comparisons was used to test for significant differences across experimental groups. Dunn's multiple comparison post hoc test was also performed. All statistics were carried out at 95% confidence intervals, therefore a significant threshold of $p < 0.05$ was used in all analyses. For Sholl analysis, repeated measures two-way ANOVA was performed using SPSS (Version 26, IBM Statistics). When comparing percentages (e.g. of regenerating cells or RGC survival after optic nerve crush), Fisher's exact test was performed between each two groups compared. $p$-values were then analyzed with the "Analyze a stack of $p$ values" function in GraphPad Prism with a Bonferroni–Dunn pairwise comparison to test for statistical significance between groups. All representative images, kymographs or micrographs were obtained from experiments which were repeated at least three times.

**Reporting summary**. Further information on research design is available in the Nature Research Reporting Summary linked to this article.

## Data availability

The RNA-sequencing datasets from peripheral DRG neurons in development and after injury[28], from RGCs during development[35] or from cultured rat primary cortical neurons[19] used for analysis in this study (Fig. 1c, d) have previously been published and deposited in NCBI Gene Expression Omnibus (accession numbers: GSE66128, GSE90654 and GSE92856, respectively). The rest of the data generated to support the findings of this study are available from the corresponding author upon reasonable request. Source data are provided with this paper.

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

## Acknowledgements

We would like to thank Prof. Joost Verhaagen for providing the viral constructs in which Protrudin was cloned, Mr Tolga Sadku for creating the Protrudin schematic diagram, the Light Microscopy Core of the NHLBI/NIH, and Mr. Raymond Fields at the National Institute of Neurological Diseases and Stroke Viral Core Facility for making the Protrudin viruses. Funding was from the UK Medical Research Council (MR/R004544/1, MR/R004463/1),; Christopher and Dana Reeve Foundation; EU ERA-NET NEURON – grant AxonRepair; Bill and Melinda Gates Foundation; International Foundation for Research in Paraplegia (IRP); Fight for Sight (5119/5120, and 5065–5066); Cambridge Eye Trust, and National Eye Research Council; the Operational Programme Research, Development and Education in the framework of the project "Centre of Reconstructive Neuroscience", Czech Ministry of Education, CZ.02.1.01/0.0./0.0/15_003/0000419; Vetenskapsrådet 2018-02124 (P.A.W); Pete Williams is supported by the Karolinska Institutet in the form of a Board of Research Faculty Funded Career Position and by St. Erik Eye Hospital philanthropic donations. Initial Protrudin constructs were made in Dr E. Reid's laboratory under the Wellcome Trust Senior Research Fellowship grant (082381); Division of Intramural Research, National Heart, Lung, and Blood Institute, NIH.

## Author contributions

V.P., C.S.P., R.E., E.R. and J.W.F. came up with the concept of the paper and designed all initial experimental procedures. V.P. performed and quantified all in vitro regeneration and validation experiments and assisted with the design and curation of in vivo data. C.S.P. and J.C. designed and performed optic nerve crush experiments. C.S.P., J.C., A.S., Y.Y., A.O., F.M.L. and R.J.W. performed the validation, execution, quantification and curation of optic nerve crush regeneration and survival data. J.R.T. and P.A.W. performed, quantified and curated all data related to the retinal explant neuroprotection experiments. K.R.M., E.R., P.A.W., H.M.G., R.E. and J.W.F. supervised the project throughout and obtained funding. V.P., R.E. and J.W.F. wrote the original draft of the paper. All authors gave comments on the manuscript and approved the final version of the paper.

## Competing interests

Authors declared no competing interests.
