## [Peer Review File · Nature Communications]

REVIEWER COMMENTS

Reviewer #1 (Remarks to the Author):

Petrova and co-workers showed a contribution of an adaptor molecule protrudin to axonal regeneration. The expression of protrudin is low in mature CNS neurons with poor regenerative capacity, and overexpression of protrudin promotes axonal regeneration after axon injury. In cultured neurons, overexpression of protrudin facilitates axonal transport of Rab11 and Integrin $\alpha 9$, resulting in accumulation of ER at the growth cone. Furthermore, protrudin overexpression exerts a neuroprotective effect on RGC neurons, and optic nerve injured-animals manifest improved regenerative capacity of protrudin-overexpressed axons. The authors conclude that protrudin facilitates axonal transport and ER accumulation at the growth cone and thereby promotes axonal regeneration after CNS injury.

The findings of this study are interesting and have the potential to be important in providing new therapies for CNS injuries. However, there are many questions about the explanation of the mechanism. It is likely that some of the protrudin mutants used in this study are not necessarily deficient in function as intended by the author and therefore may be misinterpreted.

Major comments:

1. The functional characterization of "active" protrudin is totally insufficient. Although a previous study showed that protrudin mutants lacking potential ERK phosphorylation sites showed markedly reduced affinities for Rab11 compared with the affinities of wild-type protrudin (Shirane & Nakayama, *Science* 314: 818-821, 2006), the opposite is not necessarily true. As far as I know, no one has ever proven that such a phosphomimetic variant used in this study is actually an active form. The lack of verification of actual activation by the mutant protrudin is a major problem with this study. Given that various activities have been reported for protrudin, it is necessary to specify what the activity is for. For example, does active protrudin alter the ability to bind Rab11 or modify Rab11 activity? Does it increase the ability of non-neuronal cells to form protrusions? Does it differ in the promotion of endosomal transport by protrudin? These mutant sites are the phosphorylation sites of protrudin by the ERK, which is expected to be mimic, but there is no evidence that it is the same as the actual phosphorylated state.

2. Various protrudin mutants have the following problems.

- The KIF5 mutant is expected to bind neither KIF5 nor VAP, given that the FFAT domain is also deleted. In this experiment, the extent of the deletion in protrudin should be more localized so that only KIF5 does not bind to the mutant.
- TM 1-3 mutant is likely to have abnormal conformation, rather than simply not binding to the ER membrane, given the large extent of the deletion.

3. In all experiments, only overexpression of protrudin was performed and there were no inhibition of function experiments. The authors should use knockdown and knockout techniques for protrudin to show reduced accumulation of Rab11 and integrin in the growth cone, reduced accumulation of ER, and reduced neural regenerative capacity.

4. The only examples given in this study are cortical neuron and regeneration in RGC. Is it limited to these neurons, or is it a phenomenon that is widespread and universal to the neuron of CNS and PNS?

5. In this study, RTN4 is used as a marker of smooth ER. On the other hand, however, RTN4 (also known as Nogo-A) functions as a very strong inhibitor of axonal regeneration following CNS injury. The accumulation of RTN4 in the growing cone would therefore be expected to prevent the regeneration of its axons. The authors need to address this point.

6. In Figure 6, the neuroprotective effect of wild-type or active-protrudin overexpression is shown

in the whole retina explant culture. However, it is not clear that the overexpression protects cell death in RGC neurons caused by optic nerve injury. The authors should examine the effect of protrudin overexpression on the neuronal death following optic nerve injury.

7. It is very interesting that protrudin exerts two functions, neuroprotection and axon elongation, but no data are provided to show the mechanism by which wild-type or active-protrudin overexpression inhibits neuronal death induced by CNS injury. Further analysis of the neuroprotective effect of protrudin is essential to overcome the limited novelty of this study. The authors should provide a mechanism for this.

8. Fig. 1D shows that the amount of protrudin mRNA increases after peripheral nerve injury, but this is a subtle difference. The authors need to examine whether it is really elevated at the protein level.

Minor comments:

1. In Figures 1C, 1D, S2F, S3C, 5D, 6C, and 6D, the sample size is not described in the legends.

2. In Figure 1, "(G)" is mislabeled as "(D)" in the legend.

3. In Figure S2C, the letter of "E." is misplaced in the center of the figure.

4. In Figure S2E, S2F, and 4D, the vertical axis characters are not displayed properly.

5. In Figure S2E, "There were no significant differences between the four conditions" is written in the legend. Should it be "the three conditions?"

6. In Figure S3E, the scale bar is missing.

7. In Figures 3A and S4A, the length of a scale bar is not provided in the legends.

8. In Figure S5, "(B)" is missing in the legend.

9. p. 29. The reference "Farias, 2019" is not formatted properly.

Reviewer #2 (Remarks to the Author):

Petrova and colleagues have performed a series of experiments investigating the efficacy of Protrudin in promoting axon regeneration after CNS injury. Protrudin was found to be excluded from cortical axons as they mature into neuronal circuits in a time frame coinciding with their transition to an inability to regenerate. Overexpressing wildtype protrudin or a phosphomimetic active form of protein promoted axon regeneration in an in vitro laser axotomy assay and enhanced cortical axon outgrowth in vitro. Interestingly, protrudin overexpression enhanced accumulation of endoplasmic reticulum, integrins, and Rab11 endosomes into the distal axon, and importantly, removing Protrudin's endoplasmic reticulum localization, kinesin-binding, or phosphoinositide-binding properties abrogated its regenerative effects. Finally, overexpression of wildtype or active protruding prevented RGC death in a glaucoma explant model and axon regeneration after optic nerve injury in vivo.

The Fawcett lab has made substantial contributions in the field of axon regeneration, by combining state-of-the-art cell biological techniques and questions with the injured spinal cord. The presented work is another brilliant example of this approach. This study unravels the endoplasmic reticulum as a new player driving axon regeneration. The experiments provide a novel understanding of how

the scaffolding of axonal organelles and motors into the growth cone is important for regenerative outcome. The manuscript is of high technical quality and very interesting for the general readership of Nature Communications. There are few comments that should be worked on.

MAJOR

1. The laser axotomy assay demonstrates that overexpression of wildtype protrudin or active protrudin enhances axon regeneration in mature neurons. Is there a specific time-frame post-axotomy in which Protrudin enhances outgrowth? Does it enhance initial growth cone formation or growth after growth cone formation has already occurred? The authors should compare the kinetics of axon growth in regenerating axons between the different groups and assess this with their existing data.

2. The effect of Protrudin overexpression on neuroprotection in the glaucoma model presented in Figure 6 is very exciting. It would be of significant interest to the community to assess whether Protrudin overexpression can also prevent cell loss after axotomy in vivo. Conversely, it would be interesting to know if Protrudin overexpression prevents neurodegeneration in the explant model but not in the in vivo situation, and if this were the case, it would not change the importance of this paper. Given the experiments performed in Figure 5, the authors should perform retinal whole-mount staining or stain tissue sections of the retina and quantify the extent of retinal ganglion cell loss 2 weeks after optic-nerve crush after overexpression of the wildtype protrudin and active protrudin constructs relative to control.

The reviewer is aware of the current Covid-situation. In case this is the only hurdle preventing rapid publication, the reviewer can be convinced that this should not be done. If, however, other reviewers ask for experimental additions, preventing a rapid turnaround, we would like to kindly ask that this experiment will be performed.

MINOR –

-In introduction – “leading to better regenerative after injury” should be “better regeneration”

-Later in introduction – “Interfering with other key domains of also eliminated the regenerative effects.” – missing Protrudin?

-The figure legend for Figure 1 is missing “G”. It is not clear at what timepoint DIV the analysis of the overexpressed Protrudin is done at.

-The discussion is quite long and should be re-written to be more concise.

Reviewer #3 (Remarks to the Author):

Comments to Authors:

In this study, Petrova and colleagues have utilized imaging techniques to study effects of overexpression of the ER protein protrudin in axonal regeneration. The main finding from the work is that protrudin, especially in its phosphorylated form, is able to promote axon regeneration and it requires all of its membrane targeting and protein binding domains to be able to perform this function, suggesting that this process requires the scaffolding of kinesin, Rab11 vesicles and the endoplasmic reticulum (ER). Similar findings have been reported for a role of protrudin in neurite outgrowth, but this manuscript is the first report in the context of axonal regeneration. Increasing the flux of Rab11 vesicles in axons has been previously shown to promote axonal regeneration, so given that the authors show that protrudin overexpression promotes Rab11 vesicle trafficking in axons, it is expected that protrudin would be able to promote axonal regeneration as well. The most original finding from this manuscript is the requirement of the ER localization of protrudin in

the process. This result will open new questions regarding the role of the ER in axonal regeneration. A main limitation of this study is that it is not clear how the various interactions of protrudin are functionally interconnected. This limitation limit the conceptual advance contributed by this manuscript relative to previous knowledge.

Specific comments:

- In several experiments, a phosphomimetic form of protrudin is used. The study cited regarding this construct (Shirane et al 2006) finds that protrudin gets phosphorylated upon ERK activation, that protrudin binds better to Rab11 upon ERK activation, and identifies potential ERK phosphorylation sites in protrudin. However, this study does not directly test whether phosphomimetic mutants of protrudin indeed recruit more Rab11 than WT forms of the protein. The authors should analyse whether in fact there is more Rab11 bound to the active form of protrudin when compared to WT, and whether other properties of Protrudin (such as kinesin binding or VAP binding) are affected in this mutant.
- It was shown previously that overexpression of Rab11 promotes axonal regeneration by increasing Rab11 trafficking into the axon, and the present manuscript shows that protrudin overexpression increases Rab11 trafficking in the axon. However, the authors here report that the ER connection of protrudin is also crucial for axonal regeneration, suggesting its role in tethering Rab11 to KIF5A is not enough to achieve a considerable effect in this process. Could these effects be intertwined? Is Rab11 trafficking into the axon affected by overexpression of the ER-localization defective mutants?
- In Fig 4E, the graph suggests that both protrudin mutants defective in ER binding (the construct lacking the motif FFAT and the construct lacking TM domains) are not as powerful in promoting axon regeneration as the active form of protrudin, although these mutants have some effect relative to the control. Given that these mutants can still bind kinesin and Rab11, one wonders whether the effect of protrudin in axonal regeneration is the result of these effects, independently, the ER into the growing axon. Would it be possible to promote the regeneration of axons with co-expression of a protrudin construct that can bind VAP and KIF5A but not Rab11 and a protrudin construct that can bind Rab11 and KIF5a but not VAP? In other words, is it strictly necessary to have a scaffold of the 3 components (ER, Rab11 and kinesin) or is protrudin performing two separate jobs by separately linking the ER to kinesin and Rab11 to kinesin?

Minor comments:

- Some measurements of mRNA in Fig1D do not have error bars. These are necessary to support the claim that protrudin expression is increased upon injury.

Reviewer #1 (Remarks to the Author):

*Petrova and co-workers showed a contribution of an adaptor molecule protrudin to axonal*
*regeneration. The expression of protrudin is low in mature CNS neurons with poor*
*regenerative capacity, and overexpression of protrudin promotes axonal regeneration after*
*axon injury. In cultured neurons, overexpression of protrudin facilitates axonal transport of*
*Rab11 and Integrin $\alpha 9$, resulting in accumulation of ER at the growth cone. Furthermore,*
*protrudin overexpression exerts a neuroprotective effect on RGC neurons, and optic nerve*
*injured-animals manifest improved regenerative capacity of protrudin-overexpressed axons.*
*The authors conclude that protrudin facilitates axonal transport and ER accumulation at the*
*growth cone and thereby promotes axonal regeneration after CNS injury.*

*The findings of this study are interesting and have the potential to be important in providing*
*new therapies for CNS injuries. However, there are many questions about the explanation of*
*the mechanism. It is likely that some of the protrudin mutants used in this study are not*
*necessarily deficient in function as intended by the author and therefore may be misinterpreted.*

*Major comments:*

*1. The functional characterization of “active” protrudin is totally insufficient. Although a*
*previous study showed that protrudin mutants lacking potential ERK phosphorylation sites*
*showed markedly reduced affinities for Rab11 compared with the affinities of wild-type*
*protrudin (Shirane & Nakayama, Science 314: 818-821, 2006), the opposite is not necessarily*
*true. As far as I know, no one has ever proven that such a phosphomimetic variant used in this*
*study is actually an active form. The lack of verification of actual activation by the mutant*
*protrudin is a major problem with this study. Given that various activities have been reported*
*for protrudin, it is necessary to specify what the activity is for. For example, does active*
*protrudin alter the ability to bind Rab11 or modify Rab11 activity? Does it increase the ability*
*of non-neuronal cells to form protrusions? Does it differ in the promotion of endosomal*
*transport by protrudin? These mutant sites are the phosphorylation sites of protrudin by the*
*ERK, which is expected to be mimic, but there is no evidence that it is the same as the actual*
*phosphorylated state.*

The reviewer raises several important points: is there evidence that phosphomimetic Protrudin
has additional activity, and do we understand the mechanism? Our approach was to target
Protrudin phosphorylation sites known to be phosphorylated by ERK. We, therefore, created a
phosphomimetic form of the protein by mutating specific serine and threonine residues to
aspartic acid, which is the most widely accepted approach to generate phosphomimic proteins
¹. Our intention was not to copy the actions of ERK, but to investigate whether mimicking
phosphorylation at these sites would have favourable effects for our study. There is data in the
submitted paper that demonstrates that the phosphomimetic Protrudin has additional effects to
the native molecule, and we have performed several additional experiments to assess whether
phosphorylated Protrudin is in fact “active”.

Below, we respond to each question by the reviewer covering the following effects of
phosphomimetic and wild-type Protrudin on:

**1) Rab11 binding;** The reviewer asks whether the greater efficacy of phosphomimetic
Protrudin might be due to increased Rab11 binding, because one of the sites that was
mutated is usually called the Rab11-binding site. We have performed an additional set
of experiments that address this point described as follow:

- **Co-immunoprecipitation;** We performed co-immunoprecipitation analysis in
HeLa cells co-transfected with either control GFP, wild-type or phosphorylated
Protrudin-GFP together with either wild-type, dominant negative (DN, GDP-
bound) or constitutively active (CA, GTP-bound) Rab11-RFP in order to assess
whether there is altered Rab11-binding upon mutant Protrudin overexpression, but
our findings were inconclusive (Fig. R1A-B). All co-immunoprecipitation
experiments were conducted in accordance with published protocols and previous
findings in Shirane and Nakayama, 2006 ² and we have performed at least 3
individual experiments in each condition. We found that both wild-type and
phospho-Protrudin bound to all forms of Rab11 compared to GFP control. There
was a trend for less binding of active Protrudin to wild-type Rab11, for increased
binding of active Protrudin to constitutively active Rab11 compared to wild-type
Protrudin, but there were no differences in binding to dominant negative Rab11.
Overall, we did not observe any statistically significant changes across any of the
conditions (Fig. R1C).

**Fig. R1. Co-Immunoprecipitation of Protrudin and Rab11.** A. A schematic diagram to
 describe the co-immunoprecipitation methodology. B. Immunoblots from IP and input to show
 that wild-type and also constitutively phosphorylated Protrudin bind to all forms of Rab11.
 GFP on its own, does not bind to any forms of Rab11. C. Quantification of the band density of
 Rab11 in IP after immunoblotting with anti-Rab11 antibody from several different experiments
 ($n = 3-4$), normalised to the amount of GFP overexpression (*one-way ANOVA*, $p = 0.004$ for

WT Rab11, $p = 0.01$ for DN Rab11 and $p = 0.02$ for CA Rab11). There were no differences
observed between the binding of wild-type and mutant Protrudin to any one form of Rab11.

- **Rab11 Live-cell imaging;** We also looked at Protrudin-Rab11 interactions in
cortical neurons using live cell imaging. These experiments are in Figure 3 of our
manuscript and show that either wild-type or active Protrudin overexpression
results in an increased amount of Rab11 vesicles in distal axons, however there was
no difference between expression of wild-type or active Protrudin. We did not find
any difference between wild-type and phosphomimetic Protrudin in their effects on
the directionality of Rab11 endosomal transport. We have now addressed this in the
manuscript by adding the sentence “The finding that active Protrudin has no
additional effect on axonal transport compared with wild-type Protrudin suggests
that phospho-mimetic Protrudin does not function to further stimulate Rab11
transport” on pages 13 and 14.

2) **Non-neuronal cell line protrusion formation;** As suggested by the reviewer, we
performed experiments to look at protrusion formation in HeLa cells in accordance with
previous findings by Shirane and Nakayama, 2006. We included the data generated
from this experiment as a supplementary figure in the manuscript. We found that
expression of the phosphomimetic Protrudin mutant resulted in a higher percentage of
cells forming protrusions as well as in longer protrusions compared to wild-type
Protrudin, confirming the “active” phenotype of the mutant. This data is now presented
in Fig. S1 and is described in the text on page 4.

3) **Endosomal Transport;** As mentioned above we performed live-cell imaging
experiments to assess the role of active Protrudin on endosomal vesicle transport. To
do this we co-expressed either wild-type or active Protrudin together with one of two
endosome markers which we have successfully used in the past – Rab11 and integrin
alpha 9. We observed an increase in endosomal trafficking in the distal axon, but we
did not observe any significant changes between wild-type and active Protrudin. All
data is presented in Figure 3.

In our study, we found the most pronounced effects of the phosphomimetic Protrudin over the
wild-type protein on:

1) **Axotomy;** Phosphomimetic Protrudin has a greater ability to promote axon
regeneration *in vivo* compared to wild-type and there is a similar trend *in vitro*. This is
shown in Figures 2 and 5.

2) **ER localisation;** Phosphomimetic Protrudin recruits ER to growth cones more
effectively than wild-type Protrudin, shown in Figure 4. We have now used an additional
marker to localize ER which confirms the result, shown in our response to point 4. This
is shown in Fig. S6 and described on page 17-18. We have now added the sentence
“These findings indicate that phosphomimetic, active Protrudin has stronger effect on
ER mobilisation compared to wild-type Protrudin, in contrast to its effect on Rab11
transport (Fig. 3)” to the text on page 18. We also observed increased ER labelling
(RTN4) in the protrusions of HeLa cells expressing active Protrudin, but not wild-type
Protrudin. This is in supplementary figure S1D, and is described in the text on page 17:
“We observed a similar accumulation of ER at the tip of protrusions in HeLa cells
expressing active but not wild-type Protrudin (**Fig. S1D**).” This is also discussed in the
response to points raised by reviewer #3, and may be occurring through an increased
interaction with VAP proteins (please see below).

3) **Neuroprotection;** We have performed a new experiment looking at neuroprotection 2
152 weeks after optic nerve crush which shows that phosphomimetic Protrudin is more
effective at preserving retinal ganglion cells than the wild-type protein. This data is now
included in Figure 5, and is described in the text on page 22.

Overall, we have shown that phosphomimetic Protrudin has greater effects on axon
regeneration and neuroprotection than the wild-type protein. However, we do not have
evidence that phosphomimetic Protrudin binds more strongly to Rab11. Our results therefore
do not support the possibility that the stronger effect of phosphomimetic Protrudin on
regeneration and protection is via enhanced Rab11 association. An effect via the endoplasmic
reticulum is more probable from our results. We have used the term “active Protrudin” for the
phosphomimetic form in order to aid comprehension and in accordance with our protrusion
formation results in HeLa cells, but we could change this term to phosphomimetic Protrudin if
the reviewers feel strongly.

2. *Various protrudin mutants have the following problems.*

- *The KIF5 mutant is expected to bind neither KIF5 nor VAP, given that the FFAT domain is*

*also deleted. In this experiment, the extent of the deletion in protrudin should be more*
*localized so that only KIF5 does not bind to the mutant.*

*- TM 1-3mutant is likely to have abnormal conformation, rather than simply not binding to the*
*ER membrane, given the large extent of the deletion.*

The reviewer makes an important point about the KIF5 mutant used in this paper – the large
deletion does indeed encompass both the CC domain as well as the FFAT domain which is
important for VAP-A binding. We chose to use this mutant rather than the specific CC domain
mutant because of the 2011 paper by Matsuzaki *et al.*, which meticulously dissected the
interaction domains of Protrudin and various key binding proteins. This indicates that there is
also binding between VAP-A and KIF5, and that the dual FFAT / CC domain mutant prevented
interaction with KIF5³. We used this dual mutant to robustly prevent the KIF5 interaction.

In order to achieve a more specific ablation of the interaction between Protrudin and KIF5, we
have also created a mutant in which only the coil-coiled domain (Δ CC) is deleted as suggested
by the reviewer. This mutant has previously been validated and shown to disrupt the interaction
between Protrudin and KIF5³. We intended to test the effect of this mutant on regeneration in
axotomised rat primary neurons. Unfortunately, we were unable to obtain animal embryos due
to COVID-related disruption of our animal facility and this disruption will likely be ongoing
for several months. In order to address the reviewer's comments, we have changed the text
throughout our manuscript addressing the fact that this mutant also includes an FFAT domain
deletion.

We added the following text on page 14: “We also created a mutant which lacked the FFAT
(important for VAPA binding at ER contact sites) and the coiled-coil domain which has
previously been shown to disrupt the interaction of Protrudin with the anterograde axonal motor
KIF5²⁶. We termed this mutant KIF5/VAPA mutant.” In addition, we added the following
explanation on page 18: “We showed above that a combined KIF5/VAPA mutant which
disrupts binding to KIF5 reduced the regenerative effect of active Protrudin (Fig. 3F).
Interestingly, disruption of the FFAT domain alone had a similar effect on suppressing axon
regeneration to the combined KIF5/VAPA deletion mutant, underlying the importance of
Protrudin's interaction with ER contact site protein VAPA to mediate axon regeneration. These
data demonstrate that Protrudin enables the enrichment of ER in axon growth cones, and that
this supports Protrudin's regenerative effects.”

We made the TM1-3 mutant because of previous work that robustly validated its properties in
Chang *et al.*, 2013⁴. In this study, the authors show that there were no changes in Protrudin's
binding capacity when the TM1-3 domain is deleted compared to wild-type Protrudin. They
also showed that TM1-3 unlike individual transmembrane mutations (TM1-2, TM3, etc.) is
dispersed in the cytoplasm and loses its ER tubular localisation. In subcellular fractionation,
the TM1-3 is found in the soluble fraction whereas wild-type Protrudin appears in the
membrane fraction. These results suggest that the TM1-3 domains are important for Protrudin
localisation within the endoplasmic reticulum but do not impede its ability to oligomerise. In
addition, wild-type but not TM1-3 Protrudin could rescue ER tubular defects induced by
Protrudin depletion in HeLa cells. The TM1-3 mutant has therefore been previously extensively
validated and it was shown to lose its ER localisation properties without causing abnormal
protein conformation. This paper is cited in the relevant section on page 17. Cells transfected
with the TM1-3 Protrudin mutant in our study also exhibited no abnormal Protrudin
aggregation and the protein showed diffuse, cytoplasmic localisation as expected.

*3. In all experiments, only overexpression of protrudin was performed and there were no*
*inhibition of function experiments. The authors should use knockdown and knockout techniques*
*for protrudin to show reduced accumulation of Rab11 and integrin in the growth cone, reduced*
*accumulation of ER, and reduced neural regenerative capacity.*

It would certainly be useful to know whether the very small amount of endogenous Protrudin
in CNS neurons has a function. We have performed many experiments in pursuit of this. Below
we present data to show that expression of shRNA constructs targeting Protrudin lead to a
reduction in regenerative ability. However, we have not been able to validate Protrudin
silencing reliably in cortical neurons, using either shRNA or CRISPR approaches, so we cannot
make reliable conclusions from our data.

Currently, there are no commercially available animal models of Protrudin knockout.
Previously, successful knockdown of Protrudin was performed using siRNA constructs in
RPE-1 cells (human retinal pigmented epithelial cells) and in HEK293 (from human embryonic
kidney) after 48 hours⁵ as well as in HeLa and PC12 cells^{2,6}. Protrudin has also been
previously knocked down in zebrafish using morpholinos and this resulted in developmental
impairments⁷. Knockdown of endogenous rat Protrudin in rat primary neurons has not been

documented so far. Here, we used two approaches to reduce Protrudin levels in primary rat
cortical neurons *in vitro* – commercially available shRNA against rat Protrudin as well as
custom-made sgRNA for CRISPR-Cas9 knockdown (Fig. R2 below).

We used rat primary cortical neurons transfected with 4 shRNA constructs or a combination of
all four (shRNA_A, shRNA_B, shRNA_C and shRNA_D from OriGene) to test the ability of
these shRNAs to silence endogenous Protrudin. In primary cortical neurons transfection rates
are low, so to detect lowering of Protrudin by shRNA we had to rely on antibody staining.
Comparing transfected and non-transfected cells, no significant changes were observed in the
expression of the Protrudin protein with any of the shRNA constructs. However, the Protrudin
level in these neurons is so low that staining is very close to the background level. The antibody
works, because in Protrudin-transfected neurons staining is very bright (Fig. S2 in the
manuscript). Although we could not document knockdown, we performed laser axotomies in
neurons expressing either scrambled, or shRNA_A or shRNA_D and very low regeneration
rates, lower than in control or scrambled shRNA neurons were observed.

We also performed a CRISPR experiment. Two different rat-specific sgRNA sequences
(CRISPR_1 and CRISPR_3), targeting different regions of the Protrudin gene, were designed.
A non-targeting RNA (nsgCRISPR) was used as a control. We transfected primary cortical
neurons either at 3 DIV or at 10 DIV and neurons were fixed at 15 DIV and stained for HA-
tag to detect the CRISPR-Cas9 construct and with antibodies for the Protrudin protein. By
twelve days after transfection, a small proportion of cells expressing CRISPR_1 and
CRISPR_3 constructs showed no or very little Protrudin immunostaining staining (between 0-
25% compared to control levels). The rest of the cells, however, had similar amount of
Protrudin compared to the two controls. Again, we could not obtain a reliable verification that
we had reduced the already very low Protrudin level. In the absence of confirmed silencing we
decided not to include this data in our paper.

The message of our manuscript is that Protrudin overexpression is a useful approach to
promoting axon regeneration, and our experimental data provide a new understanding of distal
axon organelles as important for regenerative success. Whilst we agree that it will be important
to define the endogenous role of Protrudin, and this is something we will continue to
investigate, we do not feel that this is necessarily within the scope of our current manuscript.

**Fig. R2** shRNA and CRISPR constructs against rat Protrudin do not result in reliable
 Protrudin knockdown in rat primary cortical neurons. A. Immunofluorescence images of
 primary cortical neurons expressing each construct. Images are taken at 40x. Scale bars are 20
 μ m. B. Bar graph to show that no significant changes were observed in protrudin staining
 intensity when overexpressing each construct ($p=0.101$, Kruskal-Wallis statistic=9.182, $n=2$).
 C. Bar graph to show the percentage of regenerating axons after laser axotomy of neurons
 overexpression scrambled control and 2 shRNAs - shRNA_A and shRNA_D. D.

Immunofluorescence images of primary cortical neurons expressing GFP alone, nsgCRISPR
control, CRISPR_1 or CRISPR_3 (green) and stained for protrudin (red). Images are taken at
40x. Scale bars are 20 μm . E. Example images of a cell transfected with CRISPR_1 which
shows complete knockdown as there is no Protrudin staining. Images were taken at 40x. Scale
bars are 20 μm . Bar graph to show the percentage of Protrudin in rat cortical neurons
overexpressing each construct compared to the average intensity of Protrudin staining in GFP-
expressing neurons and the distribution of those. Error bars represent \pm SEM.

*4. The only examples given in this study are cortical neuron and regeneration in RGC. Is it*
*limited to these neurons, or is it a phenomenon that is widespread and universal to the*
*neuron of CNS and PNS?*

We thank the reviewer for this comment. At present we have examined Protrudin's effects in
two types of CNS neurons, cortical and retinal ganglion cell neurons. Endogenous and
overexpressed Protrudin's role in axon regeneration in other types of CNS neuron and in the
PNS are certainly of great interest to explore. In our study, we specifically set out to improve
the growth and regenerative potential in CNS neurons which normally have low intrinsic
regenerative abilities.

In order to assess the involvement of Protrudin in the robust regenerative response seen in the
peripheral nervous system, we believe the best approach will be to ablate Protrudin's
expression and observe any changes in the regenerative process. As mentioned in our response
to point 3 above, currently there are no available animal models of Protrudin knockdown, and
we have been unable to make reliable knockdowns. In future work it will be interesting to
follow this question, but it is outside the range of the current study. We mention mRNA levels
in sensory neurons in the paper for interest, but we can remove this data if it is confusing and
incomplete. We have removed the following reference to the PNS in the discussion section
previously found on page 26 in order to focus on our CNS findings: "Protrudin mRNA is
expressed at low levels in cultured CNS cortical neurons compared to other abundantly
expressed proteins but is found at much higher levels in sensory neurons (**Fig. 1C-D**).
Interestingly, Protrudin's pattern of expression in the developing mouse embryo as well as in
cultured DRG neurons and after peripheral lesions contrasts with that of Cacna2d2 – a calcium
channel protein which when suppressed improves axon regeneration (**Fig. 1D**)²⁷. Protrudin's
upregulation after peripheral nerve lesion suggested pro-regenerative properties."

*5. In this study, RTN4 is used as a marker of smooth ER. On the other hand, however, RTN4*
*(also known as Nogo-A) functions as a very strong inhibitor of axonal regeneration following*
*CNS injury. The accumulation of RTN4 in the growing cone would therefore be expected to*
*prevent the regeneration of its axons. The authors need to address this point.*

This is an excellent point, Reticulon-4 (RTN4), also known as Nogo-A, on oligodendrocytes
is indeed a well-proven inhibitor of axon regeneration. However, RTN4 expressed in neurons
incorporates into the endoplasmic reticulum (ER), and this endogenous RTN4 promotes growth
and regeneration in the CNS ^{8,9}, and is present in regenerating PNS axons after sciatic nerve
injury ¹⁰. Physiologically, RTN4 and all reticulon proteins function to stabilise the shape and
structure of the endoplasmic reticulum ^{11,12}. Recent studies demonstrated that endogenous
RTN4 acts to generate and maintain the shape and the structure of ER tubules and either
overexpression or deletion of the protein resulted in misshapen and not functional ER ¹². In the
same study, the authors were only able to detect RTN4 in association with the ER inside the
cell and not at the plasma membrane in primary neurons ¹².

In our study RTN4 was not overexpressed. In our rat primary cortical neurons, RTN4 was used
as a tubular ER marker in accordance with recently published literature ¹³. We used
immunostaining for endogenous RTN4 rather than overexpression of the tagged protein as
described in the manuscript. This was obviously not sufficiently clear, so we have now
modified the results section on page 17 and stated: “To examine the effects of Protrudin
overexpression on axonal ER we analyzed the distribution of endogenous reticulon 4 (RTN4)
by immunofluorescence, which reports on ER abundance in axons ¹³.”

In addition, to add a further validation of the change in distribution of ER after Protrudin
overexpression, we overexpressed REEP5-GFP in combination with either mCherry control,
wild-type or active Protrudin and measured the amount of REEP5 in growth cones in each
condition. Our results showed that overexpression of either wild-type or active Protrudin
resulted in increased amount of REEP5 at the axon tip compared to control, with a trend of
more REEP5 with active Protrudin overexpression compared to control. This data is now
Figure S6, and the associated text is on page 18.

*6. In Figure 6, the neuroprotective effect of wild-type or active-protrudin overexpression is*
*shown in the whole retina explant culture. However, it is not clear that the overexpression*
*protects cell death in RGC neurons caused by optic nerve injury. The authors should examine*
*the effect of protrudin overexpression on the neuronal death following optic nerve injury.*

We thank the reviewer for this suggestion, and we agree that it is of great importance to show
whether Protrudin provides neuroprotection at a longer time point after optic nerve injury.
We have now done this experiment and confirmed that phosphomimetic Protrudin is strongly
neuroprotective after optic nerve crush.

The whole retina explant model showed in Fig. 6 of the original manuscript represents a model
of complete axonal transection of retinal ganglion axons as the retina is excised from the eye
and placed in an ex vivo culture system. Therefore, this model is analogous to optic nerve
axotomy as described in Pattamatta *et al.*, 2016¹⁴. In this model, approximately 50% cell death
of retinal ganglion cells is observed 3 days post axotomy which makes it a reliable model for
the identification of neuroprotective treatments. Nevertheless, we understand the limitation of
measuring neuroprotection only 3 days post axotomy.

To address this issue, we performed optic nerve crush as described in the manuscript on page
19 and calculated the survival rate of retinal ganglion cells as identified by RBPMS staining 2
350 weeks post injury. We found that active Protrudin increased the proportion of surviving retinal
ganglion cells (52%) compared to either wild-type (27%) or control (28%) suggesting that
active Protrudin has a long-lasting neuroprotective effect. This new data is included in Fig. 5C
and D and is described in the text on page 22. Additional methods have also been added on
page 41.

*7. It is very interesting that protrudin exerts two functions, neuroprotection and axon*
*elongation, but no data are provided to show the mechanism by which wild-type or active-*
*protrudin overexpression inhibits neuronal death induced by CNS injury. Further analysis of*
*the neuroprotective effect of protrudin is essential to overcome the limited novelty of this study.*
*The authors should provide a mechanism for this.*

We thank the reviewer for the excellent point – we are currently applying for funding to study
the mechanisms of Protrudin overexpression on neuroprotection. This project involves
extensive analysis in order to pinpoint the exact actions of Protrudin in the axon, at the growth
cone and in the cell body, as well as its influences on intracellular signalling pathways and
survival mechanisms. These studies are, however, very extensive and we believe out of the
scope of the current study.

In order to address some of the possible neuroprotective mechanism of Protrudin's action, we
have briefly discussed here some of the possible mechanisms and if the reviewer thinks this
greatly contributes to the depth of our paper, we can add this paragraph to our Discussion.

There are several mechanisms by which Protrudin can promote neuroprotection. The most
plausible mechanism is that overexpression of Protrudin results in improved axonal transport
of growth-promoting receptors such as TrkB, integrins, PI3K etc. Once those receptors are
transported and inserted to the growth cone, they initiate a whole myriad of intracellular
signalling cascades. These, in turn result in variety of events which can contribute to
neuroprotection such as the initiation of retrograde survival signals to the soma resulting in
altered genetic and epigenetic regulation, alteration of the activation state of specific motor
proteins which in turn potentiates these effects, post-translational modifications as well as
altered transport of organelles essential for survival after injury such as the mitochondria. In
addition, protrudin has been associated with the shaping and proper function of the
endoplasmic reticulum – proper calcium homeostasis in the ER is essential for survival after
injury so one possibility is that protrudin contributes to neuroprotection by redistributing ER
along the injured axon.

*8. Fig. 1D shows that the amount of protrudin mRNA increases after peripheral nerve injury,*
*but this is a subtle difference. The authors need to examine whether it is really elevated at the*
*protein level.*

We thank the reviewer for the great suggestion. The data we presented in Fig.1D which shows
mRNA expression data during development and after peripheral nerve injury is already
published data which is freely available¹⁵ – we did not perform peripheral nerve injury
experiments in this paper. To address this question, we searched the literature available on
proteomic studies after peripheral nerve injury and we were only able to find one peer-reviewed
paper so far.¹⁶ This study, however, focuses on using the spared nerve injury (SNI) model to
induce neuropathic pain rather than observing regenerative response. Protrudin was not
detected as one of the peripheral nerve or spinal cord enriched proteins in this study. Currently,
some of our collaborators at Boston Children's Hospital, Harvard University are working to
create a comprehensive proteomic analysis of peripheral nerve and DRGs after several different

types of peripheral injury, where regeneration is either extensive or sparse. This study is
however, temporarily stalled due to the global COVID-19 pandemic and we look forward to
studying Protrudin protein changes in these datasets once they become available. We will not
have access to the data requested until the experiments described above are completed, peer-
reviewed and published.

The dorsal root ganglion data was put in the paper for general interest. It is not our data, not
essential for the paper, and if the reviewer thinks that it is incomplete and confusing, it is
probably best to remove it. We have already removed mention of this data from the start of the
discussion section which previously read “Interestingly, Protrudin’s pattern of expression in
the developing mouse embryo as well as in cultured DRG neurons and after peripheral lesions
contrasts with that of Cacna2d2 – a calcium channel protein which when suppressed improves
axon regeneration (**Fig. 1D**)”.

*Minor Comments:*

1. *In Figures 1C, 1D, S2F, S3C, 5D, 6C, and 6D, the sample size is not described in the*
*legends.*

All sample sizes have been added in the legends and highlighted for Fig. 1C-D (page 8),
for Fig. S2F (now renamed to Fig. S3F on page 5 of the Supplementary material), Fig.
S3C (now renamed to Fig. S4C on page 7 of Supplementary), Fig. 5D (now moved to
Supplementary material in Fig. S7B on page 11 of the Supplementary) and Fig. 6C-D
(pages 27-28).

2. *In Figure 1, “(G)” is mislabeled as “(D)” in the legend.*

The change has been highlighted in the manuscript on page 8.

3. *In Figure S2C, the letter of “E.” is misplaced in the center of the figure.*

The change has been made in Fig. S2C. Figure S2 is now moved and renamed to Figure
S3 on page 5 of the Supplementary material.

4. *In Figure S2E, S2F, and 4D, the vertical axis characters are not displayed properly.*

The changes have been made in the respective figures. Fig. S2 is now moved and
renamed to Figure S3 on page 5 of the Supplementary material. Fig. 4D is on pages 20-
21.

5. *In Figure S2E, “There were no significant differences between the four conditions” is*
*written in the legend. Should it be “the three conditions?”*

The change has been highlighted in the manuscript. Figure S2 is now moved and renamed
to Figure S3 on page 5 of the Supplementary material.

6. *In Figure S3E, the scale bar is missing.*

A scale bar has been added to figure S3E. The figure has now been moved and renamed
to Figure S4E on page 7 of the Supplementary material.

7. *In Figures 3A and S4A, the length of a scale bar is not provided in the legends.*

The length of the scale bars in Figures 3A on page 16 and S4A has been added to the
legend and highlighted. Figure S4A is now moved and renamed to S5A on page 8 of the
Supplementary material.

8. *In Figure S5, “(B)” is missing in the legend.*

The figure legend in Figure S5 has been changed according to changes in the figure. The
figure is now renamed to Figure S7 on pages 10-11 of the Supplementary material.

9. *p. 29. The reference “Farias, 2019” is not formatted properly.*

We have now shortened our Discussion as per Reviewer #2 request, so this part of the
text is no longer included in the manuscript.

Reviewer #2 (Remarks to the Author):

*Petrova and colleagues have performed a series of experiments investigating the efficacy of*
*Protrudin in promoting axon regeneration after CNS injury. Protrudin was found to be*
*excluded from cortical axons as they mature into neuronal circuits in a time frame coinciding*
*with their transition to an inability to regenerate. Overexpressing wildtype protrudin or a*
*phosphomimetic active form of protein promoted axon regeneration in an in vitro laser*
*axotomy assay and enhanced cortical axon outgrowth in vitro. Interestingly, protrudin*
*overexpression enhanced accumulation of endoplasmic reticulum, integrins, and Rab11*
*endosomes into the distal axon, and importantly, removing Protrudin's endoplasmic reticulum*
*localization, kinesin-binding, or phosphoinositide-binding properties abrogated its*
*regenerative effects. Finally, overexpression of wildtype or active protruding prevented RGC*
*death in a glaucoma explant model and axon regeneration after optic nerve injury in vivo.*

*The Fawcett lab has made substantial contributions in the field of axon regeneration, by*
*combining state-of-the-art cell biological techniques and questions with the injured spinal*
*cord. The presented work is another brilliant example of this approach. This study unravels*
*the endoplasmic reticulum as a new player driving axon regeneration. The experiments provide*
*a novel understanding of how the scaffolding of axonal organelles and motors into the growth*
*cone is important for regenerative outcome. The manuscript is of high technical quality and*
*very interesting for the general readership of Nature Communications. There are few*
*comments that should be worked on.*

**MAJOR**

*1. The laser axotomy assay demonstrates that overexpression of wildtype protrudin or active*
*protrudin enhances axon regeneration in mature neurons. Is there a specific time-frame post-*
*axotomy in which Protrudin enhances outgrowth? Does it enhance initial growth cone*
*formation or growth after growth cone formation has already occurred? The authors should*
*compare the kinetics of axon growth in regenerating axons between the different groups and*
*assess this with their existing data.*

The reviewer has a very interesting suggestion. The question is partly answered in Figure 2F-
G in the original paper. These analyses show that expression of Protrudin, particularly the
phosphomimetic form, speeds the formation of a new growth cone and increases the distance
over which axons regenerate. To address this question, we used our laser *in vitro* axotomy data
and performed further analysis where we compared the kinetics of axon regeneration. We
calculated kinetics by dividing the distance regenerated by each axon by the time it took to
extend (measuring from the start of axon extension) in order to obtain the speed of axon
regeneration. We did not observe any differences in the speed of regeneration between the
different conditions (Fig. 2H in revised manuscript, and associated text is on page 10). These
results taken together with the finding that axons overexpressing either wild-type or active
Protrudin start regenerating faster than control axons (Fig. 2G in revised manuscript, and text
on page 10), allow us to propose that Protrudin mainly affects the initial decision of whether
an axon will regenerate or not (i.e. the process of growth cone formation) rather the speed of
axon elongation once growth cones are formed. We have added the text “The speed of axon
extension after growth cone initiation did not differ between the three conditions (**Fig. 2H**).
This indicates that Protrudin has its most pronounced effect on initial growth cone formation,
rather than on the axon elongation phase of regeneration” on page 10.

*2. The effect of Protrudin overexpression on neuroprotection in the glaucoma model presented*
*in Figure 6 is very exciting. It would be of significant interest to the community to assess*
*whether Protrudin overexpression can also prevent cell loss after axotomy in vivo. Conversely,*
*it would be interesting to know if Protrudin overexpression prevents neurodegeneration in the*
*explant model but not in the in vivo situation, and if this were the case, it would not change the*
*importance of this paper. Given the experiments performed in Figure 5, the authors should*
*perform retinal whole-mount staining or stain tissue sections of the retina and quantify the*
*extent of retinal ganglion cell loss 2 weeks after optic-nerve crush after overexpression of the*
*wildtype protrudin and active protrudin constructs relative to control.*

*The reviewer is aware of the current Covid-situation. In case this is the only hurdle preventing*
*rapid publication, the reviewer can be convinced that this should not be done. If, however,*
*other reviewers ask for experimental additions, preventing a rapid turnaround, we would like*
*to kindly ask that this experiment will be performed.*

We thank the reviewer for this suggestion, and we agree that it is of great importance to show
whether Protrudin provides neuroprotection at a longer time point after optic nerve injury, and
this study was under way at the time of submission. The results are now in place and show
robust neuroprotection particularly by phosphomimetic Protrudin.

We performed optic nerve crush as described in the manuscript on page 22 and calculated the
survival rate of retinal ganglion cells as identified by RBPMS staining 2 weeks post injury. We
found that active Protrudin increased the proportion of surviving retinal ganglion cells (52%)
compared to either wild-type Protrudin (27%) or GFP control (28%). These results suggest that
active Protrudin has a long-lasting effect on neuroprotection. This data is now presented in Fig.
5C and D and the associated text is on page 22 in the revised manuscript.

MINOR

- *In introduction – “leading to better regenerative after injury” should be “better*
*regeneration”*

The change was highlighted in the text on page 3.

- *Later in introduction – “Interfering with other key domains of also eliminated the*
*regenerative effects.” – missing Protrudin?*

The change was highlighted in the text on page 4.

- *The figure legend for Figure 1 is missing “G”. It is not clear at what timepoint DIV the*
*analysis of the overexpressed Protrudin is done at.*

A legend for figure 1G has been added on page 8. The timepoint at which the axon-to-
dendrite ration for overexpressed Protrudin was measured was added both in the figure
legend on page 8 as well as in the text on page 6. All changes have been highlighted in
the manuscript.

- *The discussion is quite long and should be re-written to be more concise.*

This point has been addressed in the manuscript and the discussion was shortened.

Reviewer #3 (Remarks to the Author):

Comments to Authors:

*In this study, Petrova and colleagues have utilized imaging techniques to study effects of*
*overexpression of the ER protein protrudin in axonal regeneration. The main finding from the*
*work is that protrudin, especially in its phosphorylated form, is able to promote axon*
*regeneration and it requires all of its membrane targeting and protein binding domains to be*
*able to perform this function, suggesting that this process requires the scaffolding of kinesin,*
*Rab11 vesicles and the endoplasmic reticulum (ER). Similar findings have been reported for a*
*role of protrudin in neurite outgrowth, but this manuscript is the first report in the context of*
*axonal regeneration. Increasing the flux of Rab11 vesicles in axons has been previously shown*
*to promote axonal regeneration, so given that the authors show that protrudin overexpression*
*promotes Rab11 vesicle trafficking in axons, it is expected that protrudin would be able to*
*promote axonal regeneration as well. The most original finding from this manuscript is the*
*requirement of the ER localization of protrudin in the process. This result will open new*
*questions regarding the role of the ER in axonal regeneration. A main limitation of this study*
*is that it is not clear how the various interactions of protrudin are functionally interconnected.*
*This limitation limit the conceptual advance contributed by this manuscript relative to previous*
*knowledge.*

*Specific comments:*

*- In several experiments, a phosphomimetic form of protrudin is used. The study cited*
*regarding this construct (Shirane et al 2006) finds that protrudin gets phosphorylated upon*
*ERK activation, that protrudin binds better to Rab11 upon ERK activation, and identifies*
*potential ERK phosphorylation sites in protrudin. However, this study does not directly test*
*whether phosphomimetic mutants of protrudin indeed recruit more Rab11 than WT forms of*
*the protein. The authors should analyse whether in fact there is more Rab11 bound to the active*
*form of protrudin when compared to WT, and whether other properties of Protrudin (such as*
*kinesin binding or VAP binding) are affected in this mutant.*

We thank the reviewer for this comment, which is similar to point 1 raised by reviewer #1. As
described in our response to reviewer #1, we tested whether there is stronger binding of active
Protrudin to Rab11, by co-overexpressing different forms of Rab11 – wild-type, dominant
negative or constitutively active and performing co-immunoprecipitation assays with either
wild-type or active Protrudin. The results of those experiments are described in detail in
response to Reviewer 1, comment 1, and in figure R1, but briefly, we did not find a significant
difference in binding of active Protrudin to Rab11 compared with wild-type Protrudin.

To further address the points raised here, we performed multiple immunoprecipitation assays
where we overexpressed either GFP control, wild-type GFP-Protrudin or active GFP-Protrudin
and analysed endogenous protein interactions. After immuno-isolation using anti-GFP
microbeads, we performed immunoblotting for either Rab11, VAP-A or KIF5A.
Unfortunately, in these experiments we could not detect sufficient endogenous Rab11 or KIF5
after immunoprecipitation. We did however find that active Protrudin bound more strongly to
VAP-A compared to wild-type Protrudin and control GFP (Fig. R3 below). These results
suggest that one way in which active Protrudin leads to stronger actions on regeneration and
neuroprotection compared to wild-type Protrudin could be a more robust interaction with ER
contact site proteins. and is in keeping with our findings in Fig. 4. We have added new text to
include this on page 18 in the conclusions to Fig. 4. “These findings indicate that
phosphomimetic, active protrudin has stronger effects on ER mobilization compared with wild-
type protrudin, in contrast to its effects on Rab11 transport (Fig. 3).” Whilst our finding
regarding VAP-A is keeping with our other findings, we have chosen not to include this in the
manuscript because our experiments looking at endogenous Rab11 and kinesin were
inconclusive.

Overall, the data in the manuscript, combined with our findings from Rab11 and VAP-A
pulldowns suggest that the active Protrudin construct we have generated functions to enhance
regeneration mostly through ER mediated effects. Wild-type Protrudin modestly enhances
axonal ER, whilst active Protrudin does this robustly. Our pulldown data suggest that this is
mediated by an increased binding to VAP proteins (which occurs at ER contact sites), rather
than increased binding to Rab11.

To convey this in the manuscript, we have added two new sections of text. In response to the
similar point raised by reviewer one we have added new text on pages 13 and 14 that reads:
“The finding that active Protrudin has no additional effect on axonal transport compared with

wild-type Protrudin suggests that phosphomimetic Protrudin does not function to further
stimulate Rab11 transport”. As mentioned above, we have added new text on page 18: “These
findings indicate that phosphomimetic, active Protrudin has stronger effects on ER
mobilization compared with wild-type Protrudin, in contrast to its effects on Rab11 transport
(Fig. 3).”

**Fig. R3. Immunoprecipitation of VAPA after Protrudin overexpression in HeLa cells.**
 Schematic diagram to describe the immunoprecipitation methodology. B. Immunoblots from
 IP and input to show that active Protrudin binds more strongly to VAPA than wild type
 Protrudin. C. Quantification of the band density of VAPA in IP after immunoblotting with anti-
 VAPA antibody from several different experiments ($n = 4$), normalised to the amount of GFP
 overexpression (*one-way ANOVA*, $p = 0.0001$).

- It was shown previously that overexpression of Rab11 promotes axonal regeneration by
increasing Rab11 trafficking into the axon, and the present manuscript shows that protrudin
overexpression increases Rab11 trafficking in the axon. However, the authors here report that
the ER connection of protrudin is also crucial for axonal regeneration, suggesting its role in
tethering Rab11 to KIF5A is not enough to achieve a considerable effect in this process. Could
these effects be intertwined? Is Rab11 trafficking into the axon affected by overexpression of
the ER-localization defective mutants?

The reviewer raises some very relevant points regarding the relationship between
Rab11 transport and the ER. We intended to address this as suggested, examining Rab11 axon
transport in the presence of the ER localization mutants, however we have been unable to
perform these experiments due to the continued difficulty in obtaining rat embryos from our
animal facilities since the UK lockdown and continued restrictions. One issue is that our newly
built animal facility was re-allocated as a regional coronavirus testing centre. We had a limited
number of fixed cortical neurons expressing ER mutants that we have labelled for endogenous
Rab11, but we have not seen any differences in Rab11 axonal localisation in these neurons.
The data we have presented in Fig. 3 and 4 confirm the importance of both Rab11 and ER for
the regenerative effects, but we agree that we have not been able to determine which organelle
is the most important. Our *in vivo* findings confirm that active Protrudin has the most robust
effects on regeneration and neuroprotection, whilst our *in vitro* experiments suggest that
increased ER mobilisation is responsible for these effects, rather than enhanced Rab11 binding.
We have recently acquired funding to address the role of the ER in CNS axon regeneration,
and will be investigating whether we can stimulate regeneration with other ER mobilising
strategies that function independently of a direct Rab11 interaction. It is however likely that
the ER and recycling/Rab11 endosomes are intertwined, in that increased ER may serve as a
platform for increased endosomal transport, by virtue of an increase in ER-endosome contact
sites. Alternatively, it may also be possible to increase ER by increasing endosome transport,
with ER being dragged by endosomes on motors at contact sites. In future, we will also be
investigating whether interventions that increase Rab11 axonal transport (such as EFA6
deletion) and regeneration also lead to increased ER within the axon. We are also continuing
to investigate whether our other Rab11-mobilising interventions promote regeneration in either
the spinal cord or optic nerve. This work is making good progress, but is not ready for
publication.

To address this in the manuscript, we have added some text to the discussion on page 32: “An
outstanding issue is the relative contribution of the ER and Rab11 to the regenerative effects
of protrudin and active protrudin, especially because the ER is closely linked to numerous types
of endosomes through interactions at contact sites. Further work is needed to determine if
additional interventions which increase axonal ER also lead to enhanced regeneration,
independently of a direct interaction with Rab11.”

- *In Fig 4E, the graph suggests that both protrudin mutants defective in ER binding (the*
*construct lacking the motif FFAT and the construct lacking TM domains) are not as powerful*
*in promoting axon regeneration as the active form of protrudin, although these mutants have*
*some effect relative to the control. Given that these mutants can still bind kinesin and Rab11,*
*one wonders whether the effect of protrudin in axonal regeneration is the result of these effects,*
*independently, the ER into the growing axon. Would it be possible to promote the regeneration*
*of axons with co-expression of a protrudin construct that can bind VAP and KIF5A but not and*
*a protrudin construct that can bind Rab11 and KIF5a but not VAP? In other words, is it strictly*
*necessary to have a scaffold of the 3 components (ER, Rab11 and kinesin) or is protrudin*
*performing two separate jobs by separately linking the ER to kinesin and Rab11 to kinesin?*

This point is similar to the previous one, in that it would be good to be able to tease out the
contributions of Rab11 and ER to regeneration, and the Rab11 deletion mutant would have
been ideal for determining the extent of the three components functioning as a scaffold.
Frustratingly this construct proved to be toxic in cortical neurons. We tried to dilute the toxicity
by transfecting at low levels, and whilst we had some surviving neurons, there were not enough
for reproducible experiments. The toxicity suggests that the Rab11-Protrudin interaction is
essential for cell viability, and it will be important to determine why, as this may contribute to
the neuroprotective effect we observed. In an attempt to overcome this issue, we have begun
working with an inducible neuronal cell line in addition to cortical neurons, but this work is at
a very early stage, and has been hampered by the pandemic and UK lockdown. The reviewer
makes an important point, that the ER localisation mutants have some regenerative effects
relative to the control, and so we have added to the text on page 18 “However, both deletion
mutants had moderate regenerative effects compared to the controls. Whilst these were not
statistically significant, they indicate that Protrudin may exert some effects independently of
its localization to ER contact sites.”

*Minor comments:*

- *Some measurements of mRNA in Fig1D do not have error bars. These are necessary to*
*support the claim that protrudin expression is increased upon injury.*

We thank the reviewer for this comment. We have now added sample sizes for all graphs in
Figure 1D which have been highlighted in the figure legend on page 8. Each data point consists
of at least three different values – however the changes amongst these values are very small
and when compared to the scale of the whole graph, they seem to be negligible and cannot be
displayed on the graph as the error bars are too small to be visible. For example, the values for
Cacna2d2 mRNA in cultured DRG neurons are 215 ± 18 (6 hours post plating), 120 ± 26 (12
697 hours post plating), 68 ± 9 (24 hours post plating) and 64 ± 20 (36 hours post plating)
normalised expression values but the range for the Y-axis is between 0-2000 . We can provide
a full table with the data for each data point if necessary. We consulted the GraphPad Prism
FAQ page and tried to circumvent the problem by using their three advised methods to do so,
but we could not achieve a better presentation of the data.

**References**

- 1. Chen, Z. & Cole, P. A. Synthetic approaches to protein phosphorylation. *Current*
*Opinion in Chemical Biology* vol. 28 115–122 (2015).
- 2. Shirane, M. & Nakayama, K. I. Protrudin Induces Neurite Formation by Directional
Membrane Trafficking. *Science (80-.)*. **314**, 818–821 (2006).
- 3. Matsuzaki, F., Shirane, M., Matsumoto, M. & Nakayama, K. I. Protrudin serves as an
adaptor molecule that connects KIF5 and its cargoes in vesicular transport during
process formation. *Mol. Biol. Cell* **22**, 4602–20 (2011).
- 4. Chang, J., Lee, S. & Blackstone, C. Protrudin binds atlastins and endoplasmic
reticulum-shaping proteins and regulates network formation. *Proc. Natl. Acad. Sci.*
**110**, 14954–14959 (2013).
- 5. Hong, Z. *et al.* PtdIns3P controls mTORC1 signaling through lysosomal positioning. *J*
*Cell Biol* **216**, 4217–4233 (2017).
- 6. Connell, J. W. *et al.* ESCRT-III-associated proteins and spastin inhibit protrudin-
dependent polarised membrane traffic. *Cell. Mol. Life Sci.* (2019) doi:10.1007/s00018-
019-03313-z.
- 7. Zhang, C. *et al.* Role of spastin and protrudin in neurite outgrowth. *J. Cell. Biochem.*
**113**, 2296–2307 (2012).
- 8. Pernet, V. *et al.* Neuronal Nogo-A upregulation does not contribute to ER stress-
associated apoptosis but participates in the regenerative response in the axotomized
adult retina. *Cell Death Differ.* **19**, 1096–1108 (2012).
- 9. Welte, C., Engel, S. & Stuermer, C. A. O. Upregulation of the zebrafish Nogo-A
homologue, Rtn4b, in retinal ganglion cells is functionally involved in axon
regeneration. *Neural Dev.* **10**, 6 (2015).
- 10. Hunt, D., Coffin, R. S., Prinjha, R. K., Campbell, G. & Anderson, P. N. Nogo-A
expression in the intact and injured nervous system. *Mol. Cell. Neurosci.* **24**, 1083–
1102 (2003).

- 11. Rämö, O. *et al.* NOGO-A/RTN4A and NOGO-B/RTN4B are simultaneously
expressed in epithelial, fibroblast and neuronal cells and maintain ER morphology. *Sci.*
*Rep.* **6**, 1–14 (2016).
- 12. Voeltz, G. K., Prinz, W. A., Shibata, Y., Rist, J. M. & Rapoport, T. A. A class of
membrane proteins shaping the tubular endoplasmic reticulum. *Cell* **124**, 573–586
(2006).
- 13. Fariás, G. G. *et al.* Feedback-Driven Mechanisms between Microtubules and the
Endoplasmic Reticulum Instruct Neuronal Polarity. *Neuron* (2019)
doi:10.1016/j.neuron.2019.01.030.
- 14. Pattamatta, U., McPherson, Z. & White, A. A mouse retinal explant model for use in
studying neuroprotection in glaucoma. *Exp. Eye Res.* **151**, 38–44 (2016).
- 15. Tedeschi, A. *et al.* The Calcium Channel Subunit Alpha2delta2 Suppresses Axon
Regeneration in the Adult CNS. *Neuron* **92**, 419–434 (2016).
- 16. Barry, A. M., Sondermann, J. R., Sondermann, J. H., Gomez-Varela, D. & Schmidt,
744 M. Region-Resolved Quantitative Proteome Profiling Reveals Molecular Dynamics
Associated With Chronic Pain in the PNS and Spinal Cord. *Front. Mol. Neurosci.* **11**,
259 (2018).

REVIEWERS' COMMENTS

Reviewer #1 (Remarks to the Author):

I went through the rebuttal letter and the crucial figures of the revised version. In the revised paper, the authors experimentally addressed most of my questions and the results came out clean. While it is somewhat disappointing that some key experiments were not conducted for the coronavirus pandemic, I can understand the circumstances surrounding the authors. I strongly suggest to the authors that the term "active" Protrudin should be replaced with "phosphomimetic" Protrudin to avoid confusion and misunderstanding of the readers. As for the other points, the revision has been done very well and I have no further points.

Reviewer #2 (Remarks to the Author):

The authors have done an outstanding job of responding to our suggestions and have supplied additional, convincing data. The manuscript is suitable for publication.

Reviewer #3 (Remarks to the Author):

The authors have addressed my concerns

Reviewer #1 (Remarks to the Author):

I went through the rebuttal letter and the crucial figures of the revised version. In the revised paper, the authors experimentally addressed most of my questions and the results came out clean. While it is somewhat disappointing that some key experiments were not conducted for the coronavirus pandemic, I can understand the circumstances surrounding the authors. I strongly suggest to the authors that the term "active" Protrudin should be replaced with "phosphomimetic" Protrudin to avoid confusion and misunderstanding of the readers. As for the other points, the revision has been done very well and I have no further points.

We thank the reviewer for their suggestion, and we have now changed the name of "active" Protrudin to "phosphomimetic" Protrudin throughout the text and in all display items of our manuscript.

Reviewer #2 (Remarks to the Author):

The authors have done an outstanding job of responding to our suggestions and have supplied additional, convincing data. The manuscript is suitable for publication.

Reviewer #3 (Remarks to the Author):

The authors have addressed my concerns.